# CLIENT-AGNOSTIC LEARNING AND ZERO-SHOT ADAPTATION FOR FEDERATED DOMAIN GENERALIZATION

## ABSTRACT

Federated domain generalization (federated DG) aims to learn a client-agnostic global model from various distributed source domains and generalize the model to new clients in completely unseen domains. The main challenges of federated DG are the difficulty of building the global model with local client models from different domains while keeping data private and low generalizability to test clients, where data distribution deviates from those of training clients. To solve these challenges, we present two strategies: (1) client-agnostic learning with mixed instance-global statistics and (2) zero-shot adaptation with estimated statistics. In client-agnostic learning, we first augment local features by using data distribution of other clients via global statistics in the global model's batch normalization layers. This approach allows the generation of diverse domains by mixing local and global feature statistics while keeping data private. Local models then learn client-invariant representations by applying our client-agnostic objectives with the augmented data. Next, we propose a zero-shot adapter to help the learned global model to directly bridge a large domain gap between seen and unseen clients. At inference time, the adapter mixes instance statistics of a test input with global statistics that are vulnerable to distribution shift. With the aid of the adapter, the global model improves generalizability further by reflecting test distribution. We comprehensively evaluate our methods on several benchmarks in federated DG.

## 1 INTRODUCTION

A huge amount of data is being collected every second from a wide range of IoT devices, and the data have been utilized for building robust deep learning models. Federated learning (FL) has emerged as a promising paradigm to train the model indirectly accessing the distributed data such that it reduces privacy leakage. Pioneering studies such as FedAvg (McMahan et al., 2017) and FedProx (Li et al., 2020) train each local model on its own data while keeping data private and transmit model parameters to the server for obtaining a generalized global model. The parameters from local clients are aggregated in the server, and the server parameters are broadcasted to clients. This process is iteratively performed until the global model converges to a stationary point, and user privacy is ensured by sharing aggregated parameters not data itself with other clients.

In real-world scenarios, local data are collected from various domains across clients coming from different characteristics of sensors and surrounding environments. For example, in autonomous driving tasks, each vehicle captures street views and infrastructures differently from others due to variances in camera sensors, region, and other factors. These local data deviates in terms of the distribution in feature space, inducing non-iid data across clients, denoted as domain shift (Li et al., 2021b; Jiang et al., 2021). Currently, most studies have tried to solve the issues of FL on non-iid data, especially heterogeneous label distribution (Li et al., 2020; Karimireddy et al., 2020; Wang et al., 2020), but domain shift has not been fully explored in the literature yet. Domain shift also exists between training and test clients. After federated learning, the learned FL model is deployed to new customers outside the federation, *e.g.,* new vehicles or medical centers, where data distribution is shifted from those of clients inside the federation. However, most works only focus on improving model performance of the clients participated in FL, while neglecting generalization on unseen clients. In this paper, we treat federated domain generalization (federated DG), which aims to collaboratively learn a client-agnostic federated model from various distributed source domains and generalize the learned model to new clients in unseen domains, as illustrated in Fig. 1.

Previous FL works (Li et al., 2021b; Andreux et al., 2020) also tried to solve and analyze the domain shift problem in federated learning, but they only focused on training personalized models inside the federation not building the generalized global model for new clients. To improve the generalizability of the FL model, federated DG has been explored by (Jiang et al., 2021; Wu & Gong, 2021; Liu et al., 2021; Yuan et al., 2021). However, there are some limitations such as a threat of privacy leakage (Wu & Gong, 2021; Liu et al., 2021) by sharing private information with other clients, performance limitation (Yuan et al., 2021) focusing on aggregation rather than local training, and need of multiple target data for test-time adaptation (Jiang et al., 2021). In the literature of domain generalization of centralized learning, multi-source domain generalization (Gulrajani & Lopez-Paz, 2021) has been widely used to build the generalized model using multiple source domains. In federated learning, sharing data with other clients is strictly restricted preventing from serious privacy leakage, so these generalization methods are not be applicable to federated DG.

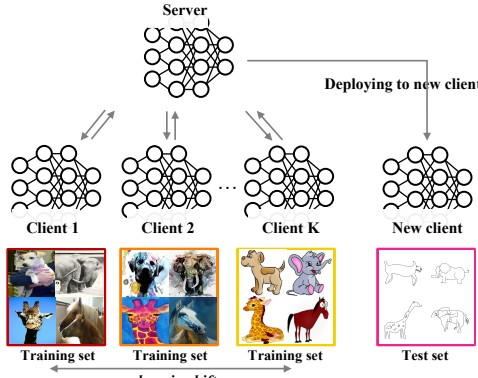

Figure 1: Federated domain generalization: a client has data belonging to a single domain that is different from other clients' domains, and the learned global model is deployed to new client.

This paper presents two approaches: (1) client-agnostic learning with mixed instance-global statistics for local training and (2) zero-shot adaptation with estimated statistics for inference. Our proposed methods, named FedIG-A, allow local models to learn client-invariant representations from other clients' data distribution while preserving privacy and let the learned global model directly generalize to unseen domains. To this end, we adopt FedBN (Li et al., 2021b) that is designed to mitigate domain shift across clients. In FedBN, local clients use local batch normalization (BN) layers and keep them local to learn client-specific representations while the remaining parts are aggregated in the server to learn client-invariant representations. However, it is difficult to explicitly train local models to learn client-invariant representations only using single domain local data. To solve the issue, we propose a novel client-agnostic learning with mixed statistics. In client-agnostic learning, we augment local features using data distribution of other clients via aggregated BN statistics from the global model, *i.e.*, global statistics. Our proposed augmentation randomly mixes instance-level and global feature statistics to produce diverse domain features. We then apply client-agnostic loss functions to learn client-invariant representations. Note that our method exploits global statistics that do not pose additional privacy leakage, the same amount with FedAvg. At inference time, we introduce a zero-shot adapter for helping the learned global model to directly bridge a large domain gap between seen and unseen clients. We mix instance statistics of a test input with global statistics that are vulnerable to distribution shift. The optimal interpolation values are different across test samples in each BN layer, thus we design the adapter for estimating the interpolation value as an instance-wise manner. With the aid of the adapter, the global model improves generalizability further by reflecting test distribution. We conduct extensive experiments on several DG benchmarks in image domain including PACS (Li et al., 2017), VLCS (Fang et al., 2013), and OfficeHome (Venkateswara et al., 2017) in the federated setting and show the effectiveness of our components.

## 2 RELATED WORKS

### 2.1 FEDERATED LEARNING

Federated learning (FL) has been extensively studied to train a global model using distributed datasets, while ensuring user privacy and reducing communication overhead issues. Most recent FL approaches focus on solving the issues of non-iid data distribution over clients, especially heterogeneous label distribution (Li et al., 2020; Karimireddy et al., 2020; Wang et al., 2020). Especially, FedProx (Li et al., 2020) incorporates a proximal term into local loss functions to regularize the local model reducing the gap with the global model. Only a few works (Li et al., 2021b; Andreux et al., 2020; Jiang et al., 2021) point out domain shift across different clients in FL. FedBN (Li et al., 2021b) and SiloBN (Andreux et al., 2020) keep BN statistics locally without aggregating them in the server for mitigating domain shift, but both methods only focus on boosting performance of clients

inside the federation. TsmoBN (Jiang et al., 2021) tackles the generalization ability of previous works. They present updating test batch normalization to adapt the global model to target clients, but it requires lots of data in target domain. Moreover, three methods do not deal with building the client-agnostic global model, *i.e.*, training local clients without domain generalization algorithms. We go beyond these approaches building the client-agnostic global model while mitigating domain shift and directly generalizing the model to new target clients indicating zero-shot adaptation.

## 2.2 DOMAIN GENERALIZATION

Domain generalization aims to train a model from multi-source or single-source domain data such that it can generalize to unseen target domains. Multi-source domain generalization (Li et al., 2019; Xu et al., 2021a; Seo et al., 2020; Zhao et al., 2021; Pandey et al., 2021; Nam & Kim, 2018; Chen et al., 2022; 2021; Lv et al., 2022) has been extensively explored to learn domain-invariant representations by minimizing domain discrepancy over multiple source domains. These methods can be applied to federated DG to help building the client-agnostic model. However, client data should be shared across clients for multi-source domain generalization, and it leads to serious privacy issues in federated learning. Single-source domain generalization (Zhou et al., 2021; Li et al., 2021a; Xu et al., 2021b; Wang et al., 2021; Carlucci et al., 2019; Huang et al., 2020; Kim et al., 2021) has tried to learn a generalized model with single source data. These algorithms can be applied to federated DG without privacy leakage. They can improve the generalization ability through domain expansion or regularization, but the performance improvement is limited since they are designed to use individual source domain data. It cannot fully exploit the advantage of federated learning. Recently, federated DG has been studied to treat the distributed multi-source domain setting. COPA (Wu & Gong, 2021) and FedDG (Liu et al., 2021) apply multi-source domain generalization methods (Li et al., 2019; Xu et al., 2021a) to the distributed setting without sharing raw data across clients, but they share classifiers and style distribution with other clients, respectively, which contain private information. Another work, CSAC (Yuan et al., 2021), proposes an aggregation method solving domain shift, but it requires a pre-training stage and achieves marginal performance improvement due to naive local training with single domain data. Compared to these works, we propose utilizing data distribution of other clients without privacy leakage for learning client-invariant representations.

## 3 CLIENT-AGNOSTIC LEARNING AND ZERO-SHOT ADAPTATION

### 3.1 PRELIMINARIES

**Notation and Problem Formulation:** Let $\mathcal{X}$ and $\mathcal{Y}$ denote the input space and the label space, respectively. $k$-th client has single domain data $D_k = \{(x_{i,k}, y_{i,k})\}_{i=1}^{n_k}$, and $\{D_1, ..., D_K\}$ is the set of distributed source domain data of $K$ clients inside the federation. In federated domain generalization (federated DG), there exists domain shift across clients, where each client data $D_k$ sampled from a domain-specific distribution $(\mathcal{X}_k, \mathcal{Y})$ different with other clients. $D_t$ indicates the target test domain data from a new client outside the federation, which distribution $(\mathcal{X}_t, \mathcal{Y})$ is shifted from that of training data. $F_\theta$ is the feature extractor parameterized by $\theta$, and $C_\phi$ is the classifier parameterized by $\phi$. Federated DG aims to learn a generalized global model $C_{\phi_G} \circ F_{\theta_G} : \mathcal{X} \rightarrow \mathcal{Y}$ by aggregating $K$ distributed clients' models $\{F_{\theta_k}, C_{\phi_k}\}_{k=1}^K$ trained on source data $\{D_k\}_{k=1}^K$, such that the global model generalizes to unseen domain $D_t$.

**Challenges:** Domain shift over clients hinders obtaining the generalized global model since the local models are easily over-fitted to their domains, indicating large model divergence across clients (Li et al., 2021b). Even though data from various domains and a large amount of data are used through federated learning, domain shifts in the distributed setting negatively affect the generalization ability both inside and outside the federation. Local models should learn domain-invariant representations such that the generalized global model is collaboratively obtained from the local models.

**FedBN:** We adopt FedBN (Li et al., 2021b) for dealing with domain shift across clients. FedBN keeps all BN layers local, and these local BN layers operate as $\gamma_k^l \frac{a^l - \mu_k^l}{\sigma_k^l} + \beta_k^l$, where $\mu_k^l$ and $\sigma_k^l$ are BN statistics of $l$-th BN layer in $k$-th client, which are calculated as running means and standard deviation. $\gamma_k^l$ and $\beta_k^l$ are learnable affine parameters, and $a^l$ indicates an input tensor. BN statistics remain local as a client-specific part while the remaining parts are uploaded and broadcasted by the

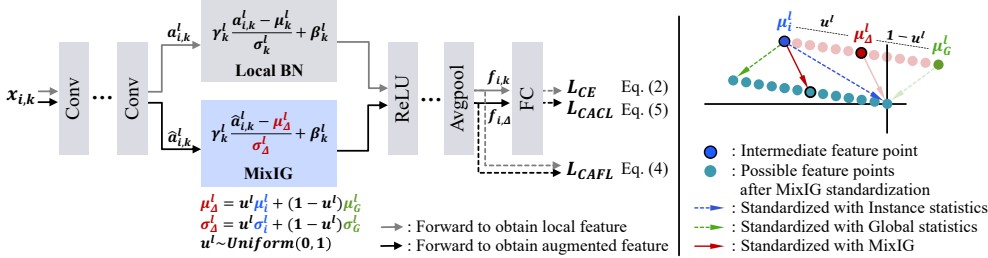

Figure 2: The local feature $f_{i,k}$ is extracted by batch statistics (Local BN), and the augmented feature $f_{i,\Delta}$ is extracted by mixed instance and global statistics (MixIG). MixIG is constructed by randomly interpolating instance and global statistics, and an intermediate feature $\hat{a}_{i,k}^l$ is standardized by MixIG (right) to possibly be various features. This operation is repeated on every BN layer. With both local and augmented features, local models learn client-invariant representations by Eq. (4) and (5).

server, denoted as a client-agnostic part. We can expect that local BN statistics learn client-specific characteristics and the client-agnostic part learns client-invariant representations. In this paper, we use $\theta = \{\theta^a, \theta^s\}$ indicating client-agnostic and client-specific parts, respectively.

In the centralized setting that accesses to multiple source datasets together, using multiple BN layers forces the model to learn domain-specific and domain-invariant characteristics separately, finding the common knowledge in multiple domains with ERM (Gulrajani & Lopez-Paz, 2021) as follows:

$$\mathcal{L}_{Centeralized\,CE} = \sum_{k=1}^{K} \frac{1}{|D_k|} \sum_{i=1}^{n_k} \text{CE}(C_{\phi_G}(F_{\{\theta_G^a, \theta_k^s\}}(x_{i,k})), y_{i,k}), \qquad (1)$$

where $\text{CE}(\cdot, \cdot)$ is the cross-entropy loss, and $\{\theta_G^a, \phi_G\}$ are shared across all domains, thus it can learn domain-invariant characteristics while $k$-th domain-specific information is learned by BN statistics $\theta_k^s$. In federated learning, local client only has single domain data different with centralized learning, thus the model cannot explicitly learn domain-invariant representations. To solve the issue, we propose to use mixed instance-global statistics in BN layers to learn domain-invariant representations.

## 3.2 CLIENT-AGNOSTIC LEARNING WITH MIXED INSTANCE-GLOBAL STATISTICS

In local training, $k$-th local model is trained with the cross-entropy loss on $k$-th dataset as follows:

$$\mathcal{L}_{CE} = \frac{1}{|D_k|} \sum_{i=1}^{n_k} \text{CE}(C_{\phi_k}(F_{\{\theta_k^a, \theta_k^s\}}(x_{i,k})), y_{i,k}). \qquad (2)$$

Before starting local training, the client receives the global model parameters $\{\theta_G^a, \phi_G\}$ and initialize the local model $\{\theta_k^a, \phi_k\}$ with the global parameters while $\theta_k^s$ remains. Then, $F_{\{\theta_k^a, \theta_k^s\}}$ and $C_{\phi_k}$ are trained on local data for long epochs. Although $\{\theta_k^a, \phi_k\}$ is initialized with the generalized global model, there is no way to learn client-invariant representations only using single domain data with the cross-entropy loss because the direct use of other clients' data leads to privacy issue. To mitigate the issue, we propose to generate diverse domain using statistics in BN layers from the global model.

**Mixed Instance and Global Statistics (MixIG):** Global statistics that are aggregated with local BN statistics in the server reflect training data distribution over clients. We exploit this property for data augmentation in local training as illustrated in Fig. 2. In naive local training, an input tensor $a_{i,k}^l$ is normalized with statistics of batch samples in $l$-th BN layer, and a local feature $f_{i,k}$ is always calculated with statistics from single domain local data. It makes the local model learn representations only in single domain. Here, we propose to normalize inputs with mixed statistics exploiting global statistics beyond local statistics. We mix mean and standard deviation of each sample with global statistics $\{\mu_G, \sigma_G\}$, *i.e.*, running mean and standard deviation, as follows:

$$\mu_\Delta^l = u^l \mu_i^l + (1 - u^l)\mu_G^l \quad \text{and} \quad \sigma_\Delta^l = u^l \sigma_i^l + (1 - u^l)\sigma_G^l, \qquad (3)$$

where $\mu_i^l$ and $\sigma_i^l$ indicate instance mean and standard deviation along the channel axis of the intermediate feature of the $i$-th input to the $l$-th BN layer, respectively. $u^l \in R^{C^l}$ is an interpolation weight vector, where each element is independently sampled from uniform distribution $U(0, 1)$ each

iteration, and $C^l$ is the feature dimension in $l$-th BN layer. The feature normalized by instance statistics contains local representative characteristics, and the normalized feature with global statistics is composed of global representations. By randomly interpolating these two statistics in all BN layers, we obtain more diverse data fully utilizing the characteristics of local and global domains. In Fig. 2, we denote the intermediate feature by $\hat{a}_{i,k}^l$ normalized with MixIG, and the augmented feature $f_{i,\Delta}$ is obtained with $\{\mu_\Delta, \sigma_\Delta\}_{l=1}^L$. We train the local model using both $f_{i,k}$ and $f_{i,\Delta}$ in a client-agnostic way, which is described in the next section. While previous works (Zhou et al., 2021; Li et al., 2021a) augment features with random noise values or styles of batch samples, our method gets access to aggregated data distribution for safe and diverse augmentations as multi-source domain generalization. It is noting that our method uses global statistics in BN layers for data augmentation, which can reduce a threat of privacy leakage different with Wu & Gong (2021); Liu et al. (2021).

**Client-agnostic Learning Objectives:** We propose a client-agnostic feature loss as follows:

$$\mathcal{L}_{CAFL} = \frac{1}{|D_k|} \sum_{i=1}^{n_k} \|f_{i,k} - f_{i,\Delta}\|_2^2. \tag{4}$$

With this loss function, the local model can explicitly extract the client-agnostic features by minimizing the distance between the original and augmented features. Here, we perturb the features on the client-specific part, *i.e.*, BN statistics, thus the client-agnostic part explicitly learns client-invariant characteristics helping to mitigate domain shift. In addition, we train the local classifier to classify the features from other domains forcing the classifier to be client-agnostic. To achieve it, the local classifier $C_{\phi_k}$ is trained with client-agnostic classification loss as follows:

$$\mathcal{L}_{CACL} = \frac{1}{|D_k|} \sum_{i=1}^{n_k} \mathrm{CE}(C_{\phi_k}(f_{i,\Delta}), y_{i,k}). \tag{5}$$

Our client-agnostic learning can be considered as a regularization method that forces local models not to deviate largely from the global model. Different with the previous work (Li et al., 2020) that directly regularizes local weight parameters with the global model, our proposed learning considers the importance of weight parameters for client-invariant representations with diverse domain data.

The overall loss for local optimization is as follows:

$$\mathcal{L}_{total} = \mathcal{L}_{CE} + \lambda_1 \cdot \mathcal{L}_{CACL} + \lambda_2 \cdot \mathcal{L}_{CAFL}, \tag{6}$$

where $\lambda_1$ and $\lambda_2$ are balancing parameters. After local training for long epochs, model parameters are aggregated by FedAvg (McMahan et al., 2017) in the server, *i.e.*, $\theta_G = \sum_{k=1}^K \frac{n_k}{n} \theta_k$ and $\phi_G = \sum_{k=1}^K \frac{n_k}{n} \phi_k$. We denote FedIG as our federated model trained by Eq. (6).

### 3.3 ZERO-SHOT ADAPTATION

At inference time, the globle model, $F_{\{\theta_G^a, \theta_G^z\}}$ and $C_{\phi_G}$, is deployed to unseen clients. It cannot generalize well to completely unseen domains, where data distribution is shifted from training distribution. In the literature of domain generalization using multiple BN layers, Seo et al. (2020) uses ensemble predictions from multiple BN layers, and Chen et al. (2022); Zhou et al. (2022) get the prediction from the selected BN that is most related to a test input. However, they cannot be applicable to the federated setting since local client model cannot access local BN layers of other clients, *i.e.*, only global BN layers are allowed to access due to user privacy. In addition, recent test-time adaptation works (Gong et al., 2022; You et al., 2021; Hu et al., 2021) use the interpolated statistics between instance and learned statistics to reflect test distribution, but their interpolation parameters are manually fixed suitable to the target domain or generated from the rule-based function containing sensitive hyper-parameters. Here, we propose to dynamically generate instance-wise interpolation parameters for mixing instance and global statistics with a learning-based network.

**Interpolated BN Statistics:** We utilize statistics of the test input with global statistics as follows:

$$\mu_t^l = \alpha^l \mu_i^l + (1-\alpha^l)\mu_G^l \quad \text{and} \quad \sigma_t^l = \alpha^l \sigma_i^l + (1-\alpha^l)\sigma_G^l, \tag{7}$$

where $\mu_i^l$ and $\sigma_i^l$ indicate instance mean and standard deviation of the input in $l$-th BN layer. $\mu_t^l$ and $\sigma_t^l$ are used for normalizing the test input tensor. $\alpha^l$ is an interpolation parameter that adjusts the contribution of instance statistics of the test sample. Ideally, optimal $\alpha$ is selected to test domains or test inputs, but we cannot access test domain. We propose a zero-shot adapter that is carefully designed to dynamically generate $\alpha$ for each input in both seen and unseen domains.

**Design of Zero-shot Adapter:** We design the zero-shot adapter $G_\varphi$ parameterized by $\varphi$, which aims to generate proper $\alpha$ for the test sample. The zero-shot adapter is separately added on each BN layer in the feature extractor. We set an input of the adapter in $l$-th BN layer as the channel-wise distance between instance and global statistics $\{\mu_i^l - \mu_G^l; \sigma_i^l - \sigma_G^l\} \in R^{2C^l}$, and an output is $\alpha^l$. With this design, the adapter estimates the statistics based on the distance between input and global statistics as an instance-wise manner at the test time.

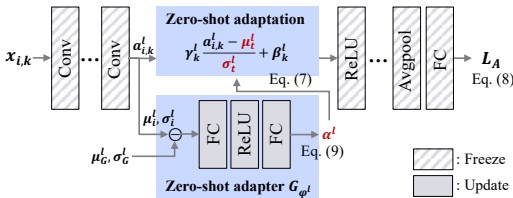

Figure 3: Zero-shot adapter takes the difference between instance and global statistics, and it generates estimated statistics by Eq. (7) and Eq. (9).

**Training Strategy for Zero-shot Adapter:** In local training, we freeze the main model, *i.e.,* $F_{\theta_k}$ and $C_{\phi_k}$, and train the adapter with the cross-entropy loss to classify the inputs as follows:

$$\mathcal{L}_A = \frac{1}{|D_k|} \sum_{i=1}^{n_k} \mathcal{L}_{CE}(C_{\phi_k}(F_{\{\theta_k^a, \theta_t^s\}}(x_{i,k})), y_{i,k}), \tag{8}$$

where $\theta_t^s$ indicates the interpolated statistics described in Eq. (7), and $\alpha^l$ is generated from $G_{\varphi^l}$, as shown in Fig. 3. To prevent the zero-shot adapter from over-fitting to each local training data, we apply reparameterization trick (Kingma & Welling, 2013). We generate $\alpha^l$ sampled from the gaussian distribution reparameterized by the zero-shot adapter as follows:

$$\alpha^l = \mathrm{T}(\delta^l z^l + \epsilon^l), \quad \text{where} \quad \delta^l, \epsilon^l = G_{\varphi^l}(\{\mu_i^l - \mu_G^l; \sigma_i^l - \sigma_G^l\}), \tag{9}$$

and $z^l$ is sampled from $N(0, 1)$. $\mathrm{T}(\cdot)$ is a clamp function to ensure $\alpha^l$ within the range of $[0, 1]$. Here, the main network and the adapter are alternately trained with the loss functions Eq. (6) and Eq. (8), respectively, freezing the other model. The training procedure for the adapter does not affect the performance on the main network since the purpose of the adapter is only to learn how to interpolate instance and global statistics for minimizing the cross-entropy loss on the trained network, which simulates the test scenario in new clients. Local zero-shot adapters are also aggregated by FedAvg.

**Zero-shot Adapter at Inference Time:** We set $\alpha^l$ as $\epsilon^l$, which is the mean of $\delta^l z^l + \epsilon^l$. The test input is normalized by interpolated statistics using $\alpha^l$ at $l$-th BN layer, and this process operates via one forward propagation same with the naive inference process. We analyze the inference cost in the experimental section. We denote our overall framework by FedIG-A.

## 4 EXPERIMENTS

### 4.1 EXPERIMENTAL SETUP

**Datasets and Settings:** We conduct extensive experiments on three DG benchmarks: PACS (Li et al., 2017), VLCS (Fang et al., 2013), and OfficeHome (Venkateswara et al., 2017). PACS contains seven categories from Photo (P), Art painting (A), Cartoon (C), and Sketch (S) domains, where domain shift is large across four domains (Dou et al., 2019; Chen et al., 2022) compared to VLCS and OfficeHome (see analysis in A.4). VLCS consists of five categories which are collected from VOC2007 (V) (Everingham et al., 2010), LabelMe (L) (Russell et al., 2008), Caltech-101 (C) (Fei-Fei et al., 2004), and SUN (S) (Xiao et al., 2010), where domain shift stems from the type of camera. OfficeHome (Venkateswara et al., 2017) contains 65 categories collected from four domains, Artistic (A), Clipart (C), Product (P), and Real world (R), and the domain shift problem is not as severe as other two datasets (Du et al., 2022; Wu & Gong, 2021). In federated DG, the client has single domain data, and there are total four clients in each DG benchmark. The global model is collaboratively learned with three clients, and the learned model is evaluated on the remaining client.

**Implementation Details:** In the federated learning process, all clients use the same architecture and hyper-parameter settings. We use ResNet-18 (He et al., 2016) pretrained on ImageNet as the backbone network. For FedIG, all batch normalization (BN) layers are replaced with two BN layers, *i.e.*, local and global BN layers, respectively. We added two fully connected layers in front of each BN layer for the zero-shot adapter. We train the network using SGD with momentum of $0.5$, the fixed learning rate of $0.01$, and the batch size of $64$ for $200$ iterations at each round. Total $40$ rounds are conducted following the federated DG setting (Yuan et al., 2021). We set $\lambda_1$ and $\lambda_2$ to $0.1$ and $4.0$ in Eq. (6), respectively. Following Li et al. (2019), we only update the classifier by Eq. (5). We conduct ablation studies about balancing parameters in A.1.

Table 1: Variants of MixIG on PACS.

| Mixing methods | Acc. |
|---|---|
| X (FedBN) | 80.79 |
| Global: $u = 0.0$ | 81.75 |
| Instance: $u = 1.0$ | 82.52 |
| Global & instance: $u = 0.5$ | 83.43 |
| **Global & instance**: $u \sim U(0, 1)$ | **84.07** |

Table 2: Ablation studies of client-agnostic learning objectives on PACS.

| Objectives | Acc. |
|---|---|
| X (FedBN) | 80.79 |
| CACL | 81.17 |
| CAFL | 82.72 |
| **CAFL+CACL** | **84.07** |

Table 3: Comparison studies to show the effectiveness our components on PACS.

| | Aug. | Loss | Inference | Acc. | | | | |
|---|---|---|---|---|---|---|---|---|
| | | | | P | A | C | S | Avg. |
| (a) | X | CE | Global statistics | 92.74 (0.97) | 77.08 (0.59) | 75.08 (0.85) | 78.28 (1.14) | 80.79 |
| | | FedProx | | 92.82 (0.90) | 76.43 (0.93) | 75.26 (0.99) | 78.95 (1.64) | 80.86 |
| (b) | Mixstyle | CAL | Global statistics | 92.04 (0.63) | 80.19 (0.28) | 77.14 (0.48) | 82.53 (0.94) | 82.98 |
| | MixIG | | | **92.99 (0.61)** | **82.17 (1.15)** | **77.71 (1.13)** | **83.40 (0.19)** | **84.07** |
| (c) | MixIG | CAL | BIN: learned alpha | 93.86 (0.53) | 78.81 (0.45) | 79.54 (0.40) | 81.94 (0.88) | 83.54 |
| | | | Ensemble BNs | 93.70 (0.39) | 83.17 (1.09) | 76.08 (1.30) | 83.14 (1.02) | 84.02 |
| | | | IABN | 93.29 (0.13) | 80.18 (1.16) | 79.38 (1.33) | 83.58 (0.91) | 84.11 |
| | | | MixNorm | 93.60 (0.52) | 82.77 (0.41) | 78.30 (1.13) | 83.12 (0.74) | 84.45 |
| | | | Random alpha | 90.67 (0.59) | 74.82 (2.97) | 74.96 (0.76) | 79.03 (1.64) | 79.87 |
| | | | Fixed alpha (0.5) | 93.59 (0.25) | 79.27 (1.40) | 79.64 (1.00) | 82.43 (1.12) | 83.73 |
| | | | Zero-shot adapter | **94.24 (0.33)** | **84.30 (0.44)** | **79.80 (1.16)** | **83.79 (0.49)** | **85.53** |

**Evaluation Protocols:** We follow the standard DG evaluation protocols in DomainBed (Gulrajani & Lopez-Paz, 2021) including dataset splits, image augmentations, and the model selection strategy. For federated DG, we set the model selection strategy as single-source DG validation on local clients and multi-source DG validation on the server. We give the details in A.2. In this protocol, we reproduce all competitive methods for a fair comparison in the federated setting. All experiments are reported the average accuracy and standard deviation over four runs with different random seeds.

## 4.2 ABLATION STUDIES OF OUR COMPONENTS

**Variants of MixIG:** We conduct experiments with our variants to deeply analyze MixIG in Table 1. Note that we use FedBN as the baseline and apply two client-agnostic learning objectives with data from variants of MixIG, but the zero-shot adapter is not used to separately analyze MixIG in this experiment. When we only use global statistics for MixIG, *i.e.*, $u = 0.0$ in Eq. (3), the input is normalized with statistics of the global model. Global statistics reflect data distribution of whole training clients, thus our objectives with normalized features allow the model to learn domain-invariant representations with global information. In the case of using instance statistics only, the style removed features (Huang & Belongie, 2017) become closer to the original features, which indicates domain-invariant information to be learned within local domain. Features normalized with global and instance statistics improve $0.96\%$ and $1.73\%$ of the average performance on PACS, respectively. By mixing global and instance statistics, the model can take advantage of both characteristics, where domain-invariant information is learned from global and local domains. Moreover, using randomly mixed statistics generates more diverse domain data such that we achieve the performance improvement further. Since global and instance statistics are randomly used for each BN layer, the model can effectively learn representations within local client, across clients, and within other clients. We also analyze the effect of distribution range in A.3.

**Effectiveness of Client-agnostic Learning:** We apply two loss functions, client-agnostic classification loss (CACL) and client-agnostic feature loss (CAFL), on augmented features. CACL builds a client-agnostic classifier, and CAFL forces the feature extractor to learn domain-invariant information by strictly minimizing the distance between two features. In Table 2, we show that each loss function is effective, and CACL and CAFL complementary operate to each other.

**Client-agnostic Learning beyond Regularization:** In Table 3-(a), we experiment the weight regularization method, FedProx (Li et al., 2020). Client-agnostic learning with the features only using global statistics can be considered as regularization not to deviate from the global model, but it even outperforms FedProx ($81.75\%$ vs $80.86\%$), here we set $\mu$ to $0.1$. Our approach regularizes the model considering the importance of model parameters suitable for client-invariant representations on local data. Beyond regularization, FedIG trains local clients with diverse domain data via MixIG.

**Effectiveness of MixIG:** MixIG is one type of feature augmentation for DG. We compare with Mixstyle (Zhou et al., 2021), which randomly mixes instance statistics within batch samples to synthesize styles and learn style-agnostic features. In Table 3-(b), we replace MixIG with Mixstyle for a fair comparison, and MixIG consistently outperforms Mixstyle on all domains. Mixstyle only

accesses the single-domain data in local training while MixIG exploits global statistics, and it shows that feature augmentation with global statistics makes the model more robust to domains.

**Comparison Studies of Inference Methods:** In Table 3-(c), we compare our zero-shot adapter with various inference algorithms. First, we compare with BIN (Nam & Kim, 2018), denoted by learned alpha, which replaces all BN layers to the weighted summation of BN and IN layers in the backbone network. In training, the learnable interpolation parameters for BN and IN are optimized, and learned parameters are used for test samples. Since the interpolation parameters are fitted to training datasets, it cannot generalize well on unseen domains. Next, we adapt various inference algorithms on the trained model using FedIG for a fair comparison. We obtain ensemble predictions from multiple local BN layers. Learned multiple BN statistics can partially reflect the distribution of unseen domains, but the performance on some domains that are severely deviated from training domains is downgraded. Furthermore, this method leads to privacy issue due to sharing multiple local statistics. IABN (Gong et al., 2022) calibrates learned statistics with instance statistics by the rule-based function when difference between learned and instance statistics is large. IABN has sensitive hyper-parameters, and these parameters should be properly selected for each domain to consistently improve the performance on all domains. Our learning based zero-shot adapter outperforms IABN on all domains without any sensitive hyperparameters. MixNorm (Hu et al., 2021) augments test input data with various spatial augmentations, here we use the original data and four augmented data, estimating test distribution more accurately. It slightly boosts the performance, but the inference time and the memory usage increase. We also compare the zero-shot adapter with naive approaches, random and fixed alpha (You et al., 2021). Compared to two results, our zero-shot adapter can generate more proper alpha values for each BN layer as an instance-wise manner. It shows that layer-wise and instance-wise generation is more effective than using the fixed alpha.

**Computational Overhead Analysis:** Table 4 shows the computational cost of each method. Training and inference times are measured as the average time per iteration with batch size 64 on an Intel Xeon Gold 6342 and a single NVIDIA RTX A5000. FedAvg, FedBN, and FedIG do not require additional processing to forward the test input at inference, while BIN consumes more time for calculating instance normalization. We also add several inference methods

Table 4: Comparison of computational cost. Acc. denotes the average accuracy on PACS.

| Methods | #Parms. | Training | Inference | Acc. |
|---|---|---|---|---|
| FedAvg | 11.31M | 1.43ms | 1.34ms | 77.35 |
| FedBN | 11.31M | 1.43ms | 1.34ms | 80.79 |
| **FedIG** | 11.31M | 2.25ms | 1.34ms | 84.07 |
| BIN w/ MixIG | 11.31M | 3.24ms | 1.35ms | 83.54 |
| FedIG w/ MixNorm | 11.31M | 2.25ms | 6.70ms | 84.45 |
| FedIG w/ IABN | 11.31M | 2.25ms | 1.57ms | 84.11 |
| **FedIG-A** | 11.53M | 3.50ms | 1.37ms | 85.53 |

(IABN, MixNorm) to FedIG to confirm the computational efficiency and performance of our FedIG-A. MixNorm forwards multiple data by batch; here, we use five for each test sample. It requires five times more memory and computations than other methods. IABN calculates instance statistics in each BN layer and gets alpha values from the rule-based function. It shows better efficiency than MixNorm, but with less performance gain. FedIG-A achieves the best accuracy by increasing marginal parameters and computations. Our FedIG and FedIG-A have a trade-off between performance and overhead, but both are superior to other baselines in terms of efficiency and performance. Users can choose the method according to the requirements of the target devices.

**Effectiveness of Zero-shot Adapter:** Using the zero-shot adapter, we plot alpha values on each layer from all test samples in unseen A domain in Fig. 4. Alpha values are different for both instance-wise and layer-wise. Since the degree of distribution shift of samples is different even in the same domain, it is effective to use different alphas for samples and layers. More results and analysis are described in A.4. In t-SNE plots, FedIG-A has fewer points in the middle, which indicate non-meaningful features, and extracts more class-discriminative features than FedIG.

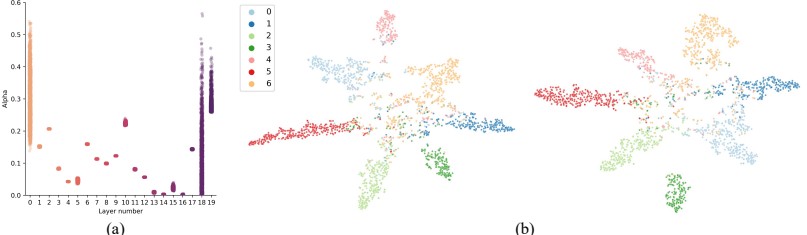

Figure 4: (a) alpha values on each layer from all test samples, and (b) t-SNE of FedIG and FedIG-A in unseen A domain, where features are normalized by global (left) and estimated statistics (right).

Table 5: Classification accuracy comparison results on PACS, VLCS, and OfficeHome. Gray color indicates methods posing privacy issues.

| Paradigm | Method | PACS | | | | | VLCS | OfficeHome |
| | | P | A | C | S | Avg. | Avg. | Avg. |
|---|---|---|---|---|---|---|---|---|
| Decentralized w/o DG | FedAvg | 90.40 (0.97) | 72.52 (2.28) | 72.59 (0.37) | 73.90 (1.74) | 77.35 | 74.86 | 63.47 |
| | FedProx | 90.49 (0.69) | 72.41 (1.06) | 73.09 (0.91) | 72.93 (1.48) | 77.23 | 74.42 | 63.02 |
| | FedBN | **92.74 (0.97)** | **77.08 (0.59)** | **75.08 (0.85)** | **78.28 (1.14)** | **80.79** | **75.33** | **64.08** |
| Decentralized w/ DG | RandAug | 93.57 (0.60) | 77.15 (1.39) | 71.68 (2.14) | 66.64 (2.34) | 77.26 | 74.32 | **64.59** |
| | Mixstyle | 92.75 (0.45) | **80.21 (0.33)** | **76.77 (0.72)** | 79.98 (2.22) | 82.43 | **75.54** | 63.24 |
| | SFA | 87.22 (1.76) | 72.01 (3.22) | 73.12 (1.03) | 79.52 (1.63) | 77.98 | 72.77 | 59.47 |
| | RandConv | 91.78 (0.40) | 77.16 (1.55) | 70.56 (3.28) | 78.78 (1.46) | 79.57 | 73.68 | 63.25 |
| | L2D | **93.59 (0.55)** | 79.69 (1.12) | 75.81 (1.37) | **82.76 (0.55)** | **82.96** | 75.37 | 63.29 |
| | JiGen | 92.14 (0.40) | 72.51 (2.00) | 72.76 (1.24) | 73.34 (1.96) | 77.69 | 75.13 | 64.09 |
| | RSC | 92.53 (0.86) | 78.10 (0.56) | **75.95 (1.08)** | **79.82 (1.31)** | **81.60** | **75.90** | 63.01 |
| | SelfReg | **93.41 (0.76)** | **78.30 (1.16)** | 74.94 (0.43) | 77.13 (2.09) | 80.94 | 72.79 | **64.85** |
| Decentralized w/ federated DG | COPA | 94.70 (1.07) | 83.75 (0.29) | 78.58 (0.96) | 84.45 (1.33) | 85.37 | 74.51 | 62.42 |
| | FedDG | 94.45 (0.44) | 83.83 (0.28) | 73.41 (1.33) | 78.40 (0.73) | 82.52 | 75.28 | 64.90 |
| | CSAC | **94.83 (0.42)** | 80.48 (1.25) | 75.46 (1.58) | 79.56 (1.35) | 82.58 | 75.21 | 64.35 |
| | **FedIG** | 92.99 (0.61) | 82.17 (1.15) | 77.71 (1.13) | 83.40 (0.19) | 84.07 | 76.62 | 64.01 |
| | **FedIG-A** | 94.24 (0.33) | **84.30 (0.44)** | **79.80 (1.16)** | 83.79 (0.49) | **85.53** | **76.68** | **64.71** |

## 4.3 COMPARISON WITH COMPETITIVE METHODS

**Competitive Methods:** In Table 5, we compare our FedIG and FedIG-A with representative methods of FL and DG. (1) Federated learning: FedAvg (McMahan et al., 2017), FedProx (Li et al., 2020), and FedBN (Li et al., 2021b); (2) Data augmentation method: RandAug (Cubuk et al., 2020); (3) Augmentation-based DG methods: Mixstyle (Zhou et al., 2021), SFA (Li et al., 2021a), RandConv (Xu et al., 2021b), and L2D (Wang et al., 2021); (4) Regularization-based DG methods: JiGen (Carlucci et al., 2019), RSC (Huang et al., 2020), and SelfReg (Kim et al., 2021); (5) Federated DG methods: COPA (Wu & Gong, 2021), FedDG (Liu et al., 2021), and CSAC (Yuan et al., 2021).

**Performance Analysis:** We first analyze the results on PACS with severe distribution shift between domains. Decentralized without DG methods achieve low performance compared to other paradigms. FedProx regularizes local models not to deviate from the global model, but it cannot solve domain shift. FedBN mitigates domain shift with local BN layers, and it can improve the performance on unseen clients. However, the performance is very limited since they do not consider learning client-invariant representations, explicitly. Decentralized with DG methods show performance improvement on several domains. Especially augmentation-based DG methods achieve significant accuracy improvement, but performance on few domains is downgraded a lot. Since augmentation-based DG methods learn domain-invariant representations only within single domain, it is not effective when augmentation on source domain does not cover test distribution. JiGen, RSC, and SelfReg consistently improve or maintain the performance on four domains compared to FedAvg, but the improvement is very marginal. COPA and FedDG allow the model to learn client-invariant representations by utilizing data information from other clients, and they obtain the highest performance on several domains among competitive methods. However, they cause serious privacy issue. Our methods, FedIG and FedIG-A, achieve the state-of-the-art performance without sharing private information across clients. FedIG outperforms all competitive methods with a large margin on most domains except for COPA. With zero-shot adaptation, we boost the performance further without significantly increasing the computational cost. On VLCS and OfficeHome, domain shift is relatively small compared to PACS. FedIG and FedIG-A consistently improve the performance almost all domains compared to other methods, indicating that our methods can be safely applied to any domains (see more results in A.5). We show the results on cross-silo FL in A.6. Moreover, we analyze the performance on clients inside the federation in A.7.

## 5 CONCLUSION

We presented client-agnostic learning with mixed instance-global statistics for local training and zero-shot adaptation with estimated statistics for inference. Our mixed instance-global statistics generate diverse domain features helping local clients to learn client-invariant representations while ensuring user privacy. In addition, our proposed zero-shot adapter directly bridges a large domain gap between training and test clients at inference time. Extensive experiments on federated DG benchmarks showed the effectiveness of our methods.

## REPRODUCIBILITY STATEMENT

Our experimental evaluation is conducted with publicly available DomainBed (Gulrajani & Lopez-Paz, 2021) and CSAC (Yuan et al., 2021). We provide the data pre-processing and hyper-parameter settings in Section 4.1 and pseudo-codes in A.8. Together with the references of related works and publicly available codes, our paper contains sufficient details to ensure reproducibility.

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

## A    APPENDIX

### A.1    PERFORMANCE DEPENDENCY OF HYPER-PARAMETERS IN OBJECTIVES

In Eq. (6), there are two hyper-parameters, $\lambda_1$ and $\lambda_2$, and we analyze performance dependency of these parameters. It is worth noting that it is an important ablation study in DG, where the DG model should generalize well on several domains, not sensitively depending on these parameters. In Fig. 5 and 6, we conduct experiments changing each parameter while fixing another parameter, $\lambda_1 = 0.1$ and $\lambda_2 = 4.0$. Although the optimal hyper-parameters are different with each domain, choosing $\lambda_1$ in $[0.1, 0.5]$ and $\lambda_2$ in $[2.0, 5.0]$ shows consistent results on all domains. In this range, FedIG-A consistently achieves high performance on all PACS domains compared to competitive methods (see Table 5). We can train and evaluate models by setting hyper-parameters to optimal values for each domain, but we set $\lambda_1$ and $\lambda_2$ to 0.1 and 4.0 for all datasets in our experiments.

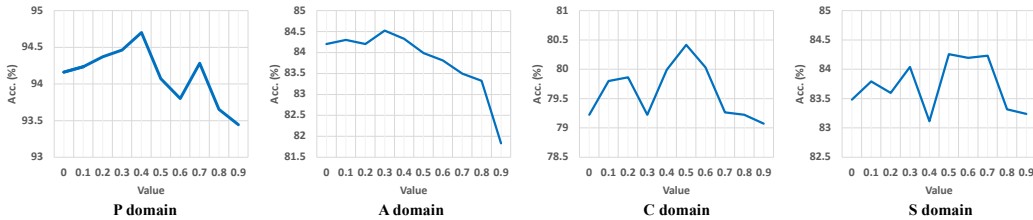

Figure 5: Ablation studies of client-agnostic classification loss on PACS. We conduct experiments using various $\lambda_1$ (x-axis) and get the performance (y-axis).

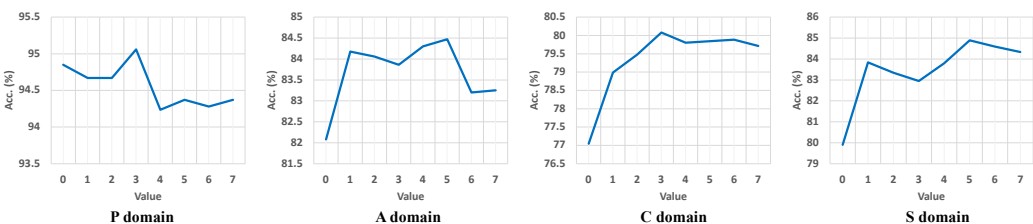

Figure 6: Ablation studies of client-agnostic feature loss on PACS. We conduct experiments using various $\lambda_2$ (x-axis) and get the performance (y-axis).

### A.2    EVALUATION PROTOCOLS FOR FEDERATED DG

We use training-domain validation set for the model selection in Gulrajani & Lopez-Paz (2021). We adopt different selection strategy for the client side and the server side to obtain the best global model in the federated DG setting. In local training, the client model is uploaded to the server when the validation accuracy on its single domain is best within 200 iterations (validation per 20 iterations). In the server, the best round model is selected when the average of the validation performance on seen clients is maximized among 40 rounds (validation per 1 round). The server can obtain the validation performance from clients using the aggregated server parameters, thus it can be possible to use the average of the validation performance on seen clients. It is practical and effective to use single-source DG validation on local training and multi-source DG validation on the server.

### A.3    MORE STUDIES ON MIXIG

In Table 6, we conduct experiments using different range of uniform distribution for mixing instance and global statistics in Eq. (3). The optimal range of augmentation is different for each test domain. In other words, the desired strength or type of augmentation is different for each domain, as shown in Table. 5. Different with augmentation-based DG methods, our method interpolates instance and global information. The performance is consistently improved in all domains due to safe and diverse augmentation using features normalized within instance and global statistics. With extrapolation

$U(-0.1, 1.1)$, the performance on S domain, which is severely deviated with P, A, and C domains, is largely improved. Using extrapolated statistics generates more diverse samples compared to using interpolated statistics, but it induces the performance drop on not severely different domains.

Table 6: Accuracy on PACS using various range of distribution for MixIG.

| Distribution | PACS | | | | | VLCS |
| | P | A | C | S | Avg. | Avg. |
| --- | --- | --- | --- | --- | --- | --- |
| $U(0, 1)$ | 94.24 (0.33) | 84.30 (0.44) | 79.80 (1.16) | 83.79 (0.49) | 85.53 | 76.68 |
| $U(0.0, 0.5)$ | 94.46 (0.21) | 84.50 (0.17) | 80.97 (1.15) | 84.46 (0.09) | 86.10 | 76.42 |
| $U(0.5, 1.0)$ | 94.58 (0.21) | 82.89 (0.59) | 78.14 (1.48) | 83.18 (0.04) | 84.69 | 76.36 |
| $U(-0.1, 1.1)$ | 94.46 (0.64) | 83.64 (0.83) | 80.12 (0.36) | 84.59 (0.05) | 85.70 | 76.46 |

### A.4 ANALYSIS OF ALPHA VALUES FROM ZERO-SHOT ADAPTER

At the test time, we use alpha values in Eq. (7) from the trained zero-shot adapter for inference. We plot alpha values on each layer from all test samples in PACS, VLCS, and OfficeHome in Fig. 7, 8, and 9. In PACS, alpha values in P, A, and C domains have similar distribution on each layer. In the low-level layers, different alpha values are used for test samples between $0.1$ and $0.6$, *i.e.*, using different amount of instance statistics, and in the high-level layers, alpha values are within $0.0$ and $0.6$. In S domain, alpha values at the low-level layers are lower than other domains. Since there is a large domain gap between S and other domains, distribution of alpha values are different. Interestingly, alpha values in the middle-level layers are almost same across test samples, *e.g.*, almost all test samples get $0.2$ on the $9$-th BN layer in P domain. Alpha values in VLCS are all between $0.0$ and $0.2$, and it shows that global statistics work well without instance statistics because domain shift existing in VLCS is not large compared to PACS. Similarly, alpha values in OfficeHome are smaller than those in PACS. The distribution of alpha values is similar across domains because domain shift is not large. The degree of shift between domains can be inferred from the distribution of alpha from the zero-shot adapter.

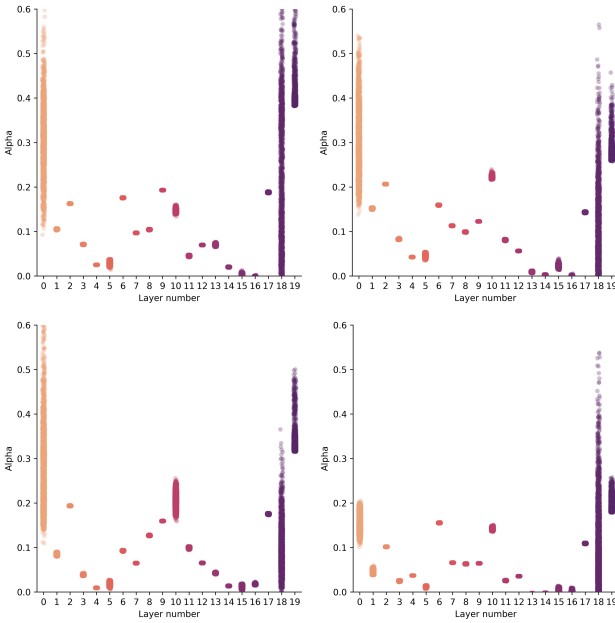

Figure 7: Alpha values on each layer from test samples in P (upper left), A (upper right), C (lower left), and S (lower right).

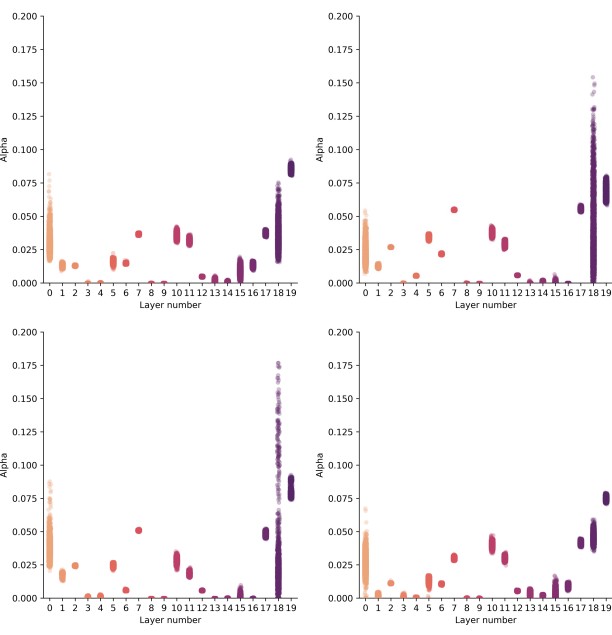

Figure 8: Alpha values on each layer from test samples in V (upper left), L (upper right), C (lower left), and S (lower right).

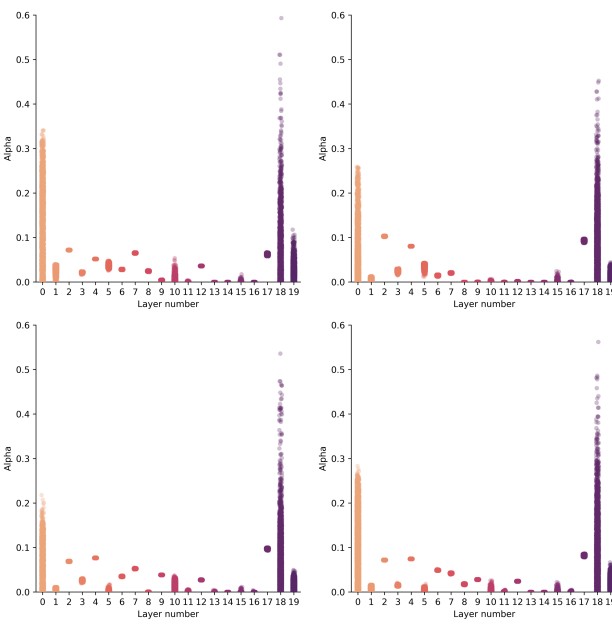

Figure 9: Alpha values on each layer from test samples in A (upper left), C (upper right), P (lower left), and R (lower right).

## A.5 EXPERIMENTAL RESULTS ON VLCS AND OFFICEHOME

In Table 7, our FedIG and FedIG-A achieve the state-of-the-art performance on VLCS. Several methods downgrade the performance on V or S domains compared with FedAvg, *e.g.*, CSAC achieves low performance on V and S domains, but we can get consistent better performance on all domains. In Table 8, both decentralized with DG and federated DG paradigms cannot get a large improvement on OfficeHome because domain shift across clients is very small. FedIG-A gets comparable results with the state-of-the-art previous methods.

Table 7: Comparison results on VLCS. Gray color indicates methods posing privacy issues.

| Paradigm | Method | VLCS | | | | |
|---|---|---|---|---|---|---|
| | | V | L | C | S | Avg. |
| Decentralized w/o DG | FedAvg | **73.53 (0.87)** | 57.92 (1.08) | 96.22 (0.93) | 71.77 (1.15) | 74.86 |
| | FedProx | 73.04 (0.53) | 57.85 (1.27) | 95.97 (0.75) | 70.83 (3.91) | 74.42 |
| | FedBN | 71.78 (0.98) | **58.71 (1.14)** | **96.67 (0.53)** | **74.15 (1.59)** | **75.33** |
| Decentralized w/ DG | RandAug | 72.42 (0.78) | 58.32 (0.74) | 95.79 (0.86) | 70.76 (2.27) | 74.32 |
| | Mixstyle | 72.61 (0.66) | 58.52 (0.66) | 97.69 (0.51) | **73.33 (1.37)** | **75.54** |
| | SFA | 65.08 (0.81) | **61.55 (1.22)** | 96.29 (1.19) | 68.15 (1.07) | 72.77 |
| | RandConv | 70.83 (0.90) | 57.16 (1.50) | 95.64 (0.23) | 71.08 (2.24) | 73.68 |
| | L2D | **72.84 (1.59)** | 59.78 (0.69) | **97.97 (0.54)** | 70.89 (1.83) | 75.37 |
| | JiGen | 73.65 (0.57) | 58.09 (0.69) | **98.06 (0.35)** | 70.72 (1.21) | 75.13 |
| | RSC | **75.27 (1.00)** | 59.79 (1.22) | 97.01 (1.01) | 71.51 (1.00) | **75.90** |
| | SelfReg | 68.13 (0.76) | **60.37 (0.66)** | 88.64 (3.09) | **74.03 (0.19)** | 72.79 |
| Decentralized w/ federated DG | COPA | 71.50 (1.05) | 61.00 (0.89) | 93.83 (0.41) | 71.72 (0.74) | 74.51 |
| | FedDG | 71.05 (0.62) | 59.46 (1.08) | 96.64 (0.86) | 73.96 (0.74) | 75.28 |
| | CSAC | 72.96 (0.91) | **59.78 (0.84)** | 96.52 (0.38) | 71.60 (1.20) | 75.21 |
| | FedIG | 73.69 (0.90) | 59.14 (0.74) | **98.00 (0.29)** | **75.65 (1.00)** | 76.62 |
| | FedIG-A | **73.93 (1.14)** | 59.29 (0.28) | 97.84 (0.37) | 75.64 (0.86) | **76.68** |

Table 8: Comparison results on OfficeHome. Gray color indicates methods posing privacy issues.

| Paradigm | Method | OfficeHome | | | | |
|---|---|---|---|---|---|---|
| | | A | C | P | R | Avg. |
| Decentralized w/o DG | FedAvg | 56.82 (0.31) | 50.53 (0.60) | 72.38 (0.24) | **74.16 (0.38)** | 63.47 |
| | FedProx | 56.58 (0.63) | 49.44 (0.39) | 72.15 (0.30) | 73.90 (0.46) | 63.02 |
| | FedBN | **58.29 (0.70)** | **51.16 (0.48)** | **72.80 (0.59)** | 74.06 (0.51) | **64.08** |
| Decentralized w/ DG | RandAug | **58.50 (0.34)** | 52.18 (0.77) | **73.17 (0.36)** | **74.52 (0.52)** | **64.59** |
| | Mixstyle | 55.57 (1.24) | 53.31 (0.65) | 70.90 (0.81) | 73.18 (0.27) | 63.24 |
| | SFA | 50.99 (1.20) | 50.97 (0.35) | 66.84 (0.65) | 69.08 (0.23) | 59.47 |
| | RandConv | 56.19 (0.80) | 53.20 (0.47) | 71.66 (0.57) | 71.94 (0.46) | 63.25 |
| | L2D | 54.70 (1.43) | **56.36 (0.28)** | 69.96 (0.85) | 72.13 (0.35) | 63.29 |
| | JiGen | 58.20 (0.67) | 50.00 (0.02) | **73.99 (0.56)** | 74.18 (0.00) | 64.09 |
| | RSC | 56.67 (0.59) | 49.59 (1.05) | 71.61 (0.47) | 74.16 (0.49) | 63.01 |
| | SelfReg | **59.26 (0.26)** | **51.84 (0.86)** | 73.46 (0.18) | **74.85 (0.62)** | **64.85** |
| Decentralized w/ federated DG | COPA | 53.39 (0.16) | 57.46 (0.47) | 68.61 (0.15) | 70.24 (0.38) | 62.42 |
| | FedDG | 59.87 (0.06) | 53.51 (0.31) | 72.81 (0.89) | 73.41 (0.41) | 64.90 |
| | CSAC | **58.97 (1.13)** | 51.61 (0.26) | **72.57 (0.18)** | 74.25 (0.49) | 64.35 |
| | FedIG | 57.68 (0.64) | 52.90 (0.13) | 71.79 (0.47) | 73.68 (0.39) | 64.01 |
| | FedIG-A | 58.62 (0.51) | **53.47 (0.21)** | 72.34 (0.45) | **74.39 (0.17)** | **64.71** |

## A.6 EXPERIMENTS ON CROSS-SILO FL

**Experimental Setup:** We expand 3 training clients to 30 clients by distributing each dataset to 10 clients on PACS. It makes the client have the small amount of data. We consider both iid and non-iid label distribution across clients. We equally distribute the data to 10 clients for iid label distribution while we use Dirichlet distribution ($\alpha = 0.5$ and $\alpha = 0.1$) for non-iid label distribution. We illustrated class distribution of clients in Fig. 10. In this setup, we randomly select 10 clients for local training with 20 iterations on each round, and total 120 rounds are conducted.

**Experimental Results:** In Table 9, 10, and 11, we compare our FedIG-A with FedProx Li et al. (2020) and FedBN (Li et al., 2021b). Our method works well when the number of clients is large. The performance is slightly downgraded in non-iid label distribution compared to the performance in iid label distribution, but we can expect that this negative effect is alleviated if the method for non-iid label distribution is added together. In the case of severe non-iid label distribution, FedIG-A significantly improves the performance compared to the baselines as shown in Table 11. It demonstrates that our method effectively solves the domain shift problem even severe label shift exists.

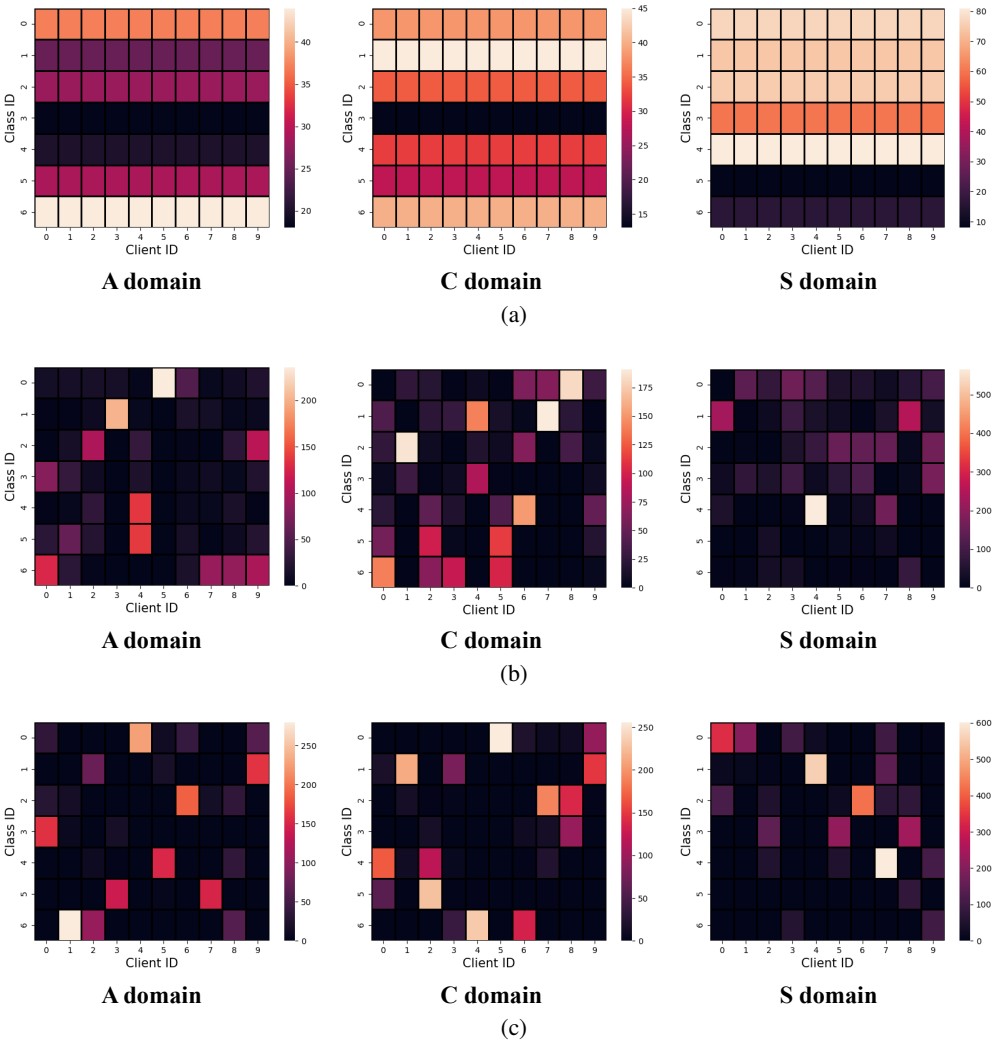

Figure 10: We plot the data distribution of clients using (a) iid data partition, (b) non-iid data partition following Dirichlet distribution of $\alpha = 0.5$, and (c) non-iid data partition following Dirichlet distribution of $\alpha = 0.1$ when A, C, and S domains are source and P is target. The color bar denotes the number of data samples, and x-axis indicates client ID and y-axis indicates class ID. Each rectangle represents the number of data samples of a specific class in a client. In this setup, 30 clients are participated in FL from A, C, and S domains, and we test the FL model on P domain.

Table 9: Performance on iid label distribution with 30 training clients.

| Method | PACS | | | | |
|---|---|---|---|---|---|
| | P | A | C | S | Avg. |
| FedProx | **94.95 (0.42)** | 73.97 (0.48) | 70.72 (0.28) | 73.76 (0.24) | 78.35 |
| FedBN | 94.53 (0.85) | 75.95 (0.17) | 70.53 (2.03) | 78.45 (2.16) | 79.86 |
| FedIG-A | 94.76 (1.14) | **84.20 (0.31)** | **79.20 (2.02)** | **82.87 (1.76)** | **85.26** |

Table 10: Performance on non-iid label distribution ($\alpha = 0.5$) with 30 training clients.

| Method | PACS | | | | |
|---|---|---|---|---|---|
| | P | A | C | S | Avg. |
| FedProx | 92.25 (0.25) | 71.75 (3.56) | 74.87 (0.61) | 69.30 (6.14) | 77.04 |
| FedBN | 92.82 (0.13) | 74.78 (1.55) | 75.50 (0.95) | 71.71 (2.66) | 78.70 |
| FedIG-A | **93.62 (0.30)** | **79.54 (0.83)** | **82.02 (0.15)** | **77.49 (1.17)** | **83.17** |

Table 11: Performance on non-iid label distribution ($\alpha = 0.1$) with 30 training clients.

| Method | PACS | | | | |
|---|---|---|---|---|---|
| | P | A | C | S | Avg. |
| FedProx | 83.17 (0.42) | 62.65 (4.42) | 66.95 (0.95) | 55.34 (4.33) | 67.03 |
| FedBN | 84.41 (1.40) | 61.62 (0.48) | 69.05 (3.81) | 59.85 (4.00) | 68.73 |
| FedIG-A | **92.13 (0.89)** | **73.29 (2.90)** | **73.19 (1.12)** | **71.96 (0.34)** | **77.64** |

## A.7 PERFORMANCE ON CLIENTS INSIDE THE FEDERATION

We measure the performance on clients inside the federation, *i.e.*, personalized performance, in Table 12. In PACS with a large domain shift across training clients, the model with local statistics, *i.e.*, the client-specific part, achieves the best performance compared to the model with global statistics, *i.e.*, the global model. Global statistics reflect data distribution of all clients, but it causes the performance degradation on each test domain, where data distribution is shifted from other clients. Our proposed zero-shot adapter can reduce the performance gap between using local and global statistics with the aid of instance statistics of the test input. In the case of VLCS and OfficeHome, global statistics help the model to generalize well to each test domain. Since the distribution shift across clients is not large, global statistics represent more general distribution compared to local statistics. It improves the model performance a lot. The zero-shot adapter also helps the model to generalize well on seen domains in VLCS and OfficeHome.

Table 12: Personalized performance on PACS, VLCS, and OfficeHome.

| Method | PACS | | | | |
|---|---|---|---|---|---|
| | P | A | C | S | Avg. |
| FedIG (local statistics) | **97.85 (0.89)** | **93.69 (2.06)** | **94.92 (0.91)** | **95.51 (0.37)** | **95.49** |
| FedIG (global statistics) | 96.86 (0.95) | 92.46 (1.79) | 93.81 (1.04) | 95.02 (0.40) | 94.53 |
| FedIG-A (global statistics) | 97.40 (0.79) | 93.53 (1.77) | 94.53 (1.08) | 95.33 (0.44) | 95.20 |

| Method | VLCS | | | | |
|---|---|---|---|---|---|
| | V | L | C | S | Avg. |
| FedIG (local statistics) | 83.79 (1.86) | **71.72 (0.84)** | 99.44 (0.41) | 80.76 (1.15) | 83.93 |
| FedIG (global statistics) | 84.80 (1.74) | 70.12 (1.24) | 99.80 (0.23) | 81.90 (1.62) | 84.15 |
| FedIG-A (global statistics) | **84.92 (1.80)** | 70.29 (1.37) | **99.85 (0.18)** | **82.09 (1.54)** | **84.29** |

| Method | OfficeHome | | | | |
|---|---|---|---|---|---|
| | A | C | P | R | Avg. |
| FedIG (local statistics) | 66.57 (1.68) | 76.11 (1.21) | 87.81 (1.32) | 79.78 (1.80) | 77.57 |
| FedIG (global statistics) | 68.05 (2.03) | **76.77 (1.20)** | **88.19 (1.14)** | 80.53 (1.91) | 78.38 |
| FedIG-A (global statistics) | **68.50 (1.82)** | 76.63 (1.16) | 88.08 (1.25) | **80.70 (1.99)** | **78.48** |

## A.8   PSEUDO-CODE FOR REPRODUCIBILITY

We describe pseudo-codes of client-agnostic learning and zero-shot adaptation in Table 13 and 14. Note that federated DG benchmarks only contain four clients, *i.e.*, three for training and one for test, thus all three clients participate in the federation at every round.

Table 13: Pseudo-code for FedIG-A training.

---

Global weights $\{\theta_G^a, \theta_G^s\}$, $\phi_G$, $\varphi_G$, Local clients' weights $\{\theta_k^a, \theta_k^s\}_{k=1}^K$, $\{\phi_k\}_{k=1}^K$, $\{\varphi_k\}_{k=1}^K$, Total round $T_{max}$, Total local iteration $E_{max}$;

---

**Server executes:**
       initialize $\{\theta_G^a, \theta_G^s\}$, $\phi_G$, $\varphi_G$;
       **for** each round $t = 1, ..., T_{max}$ **do**
           $S_t \leftarrow$ (random set of m clients);
           **for** each client $k \in S_t$ **in parallel do**
               **Load** $\{\theta_k^a, \theta_k^s\}$, $\phi_k$, $\varphi_k \leftarrow$ LocalUpdate($k$, $\{\theta_G^a, \theta_G^s\}$, $\phi_G$, $\varphi_G$);
           **Update** global weights $\{\theta_G^a, \theta_G^s\}$, $\phi_G$, $\varphi_G$ by FedAvg (McMahan et al., 2017);
       **Output**: $\{\theta_G^a, \theta_G^s\}$, $\phi_G$, $\varphi_G$.

**function** LocalUpdate($k$, $\{\theta_G^a, \theta_G^s\}$, $\phi_G$, $\varphi_G$): // Run on client k
       **Load** $\theta_k^a$, $\phi_k$, $\varphi_k \leftarrow \theta_G^a$, $\phi_G$, $\varphi_G$;
       **for** each local iteration $i = 1, ..., E_{max}$ **do**
           **Shuffle** training set $D_k$;
           **Fetch** mini batch $\{x_{i,k}, y_{i,k}\}_{i=1}^n$ from $D_k$;

           // Train the main network
           **Obtain** loss $\mathcal{L}_{CE}$ by Eq. (2);
           **Obtain** augmented features $\{f_{i,\Delta}\}_{i=1}^n$ with $\theta_G^s$ by Eq. (3);
           **Obtain** loss $\mathcal{L}_{CACL}$, $\mathcal{L}_{CAFL}$ by Eq. (4) and (5);
           **Update** local weights $\{\theta_k^a, \theta_k^s\}$, $\phi_k$ by minimizing Eq. (6);

           // Train the zero-shot adapter
           **Obtain** features with estimated statistics by Eq. (7) and (9);
           **Update** local weights $\varphi_k$ by minimizing Eq. (8);

       **Output**: $\{\theta_k^a, \theta_k^s\}$, $\phi_k$, $\varphi_k$.

---

Table 14: Pseudo-code for FedIG-A inference.

---

Global weights $\{\theta_G^a, \theta_G^s\}$, $\phi_G$, $\varphi_G$, Test client's weights $\{\theta_t^a, \theta_t^s\}$, $\phi_t$, $\varphi_t$;

---

**Deploy to test client:**
       **Load** $\{\theta_t^a, \theta_t^s\}$, $\phi_t$, $\varphi_t \leftarrow \{\theta_G^a, \theta_G^s\}$, $\phi_G$, $\varphi_G$;
       **Obtain** test set $D_t$;
       **for** each test forward $i = 1, ..., n_t$ **do**
           **Fetch** $x_{i,t}$ from $D_t$;
           **Obtain** $f_{i,t}$ by Eq. (7) and (9);
           **Obtain** the prediction with $C_{\phi_t}$;

---

