# OpenReview forum: "Client-agnostic Learning and Zero-shot Adaptation for Federated Domain Generalization"
_ICLR.cc/2023/Conference — Submitted to ICLR 2023_

### Official Review · Reviewer_LXCP · 2022-10-19

**Confidence:** 4
**Correctness:** 3
**Technical Novelty And Significance:** 3
**Empirical Novelty And Significance:** 3
**Recommendation:** 6

**Clarity, Quality, Novelty And Reproducibility:**

Clarity:
This paper is well-writen and clear.

Quality:
The related work is conprehensive, the methods are illustrated well. But the experiment setting is limited as mentioned in weakness.

Originality:
As far as I know, the proposed method is novel. There is no other works that consider mixing the local and global BN statistics and the zero-shot adaptation.

**Strength And Weaknesses:**


Strengths
1. The problem that this paper aims to solve is important for the real-world FL.
2. The related work section well reviews the federated domain generalization and the problems of directly using conventional domain generalization methods into federated domain generalization. Especially the differences between the multi-source and single-source methods can well motivate the necessity of developing multi-source domain generalization methods with privacy protection.
3. This paper has considered the privacy problem and the proposed technique does not introduce extra privacy leakage than FedAvg. The motivations of the proposed methods are clear. And the method shows some novelty.

Weaknesses
1. It seems that the client-invariant features are learned based on mixing local and global statistics. However, the global statistics do not include the un-seen clients. How to ensure the claimed client-invariant features can generalize well on un-seen clients.
2. The experiment setting is limited. Only four clients are considered. The cross-silo FL may have much more clients.
3. There is no theoretical analysis of the generalization performance.

**Summary Of The Paper:**

Federated domain generalization aims to learn a global model from various distributed source domains and generalize the learned model in completely unseen domains, while keeping the privacy of the source domains. This paper proposes client-agnostic learning with mixed instance-global statistics and zero-shot adaptation with estimated statistics to implement federated domain generalization. This paper augments local features via the global statistics in the global model's batch normalization layer. And local models learn client-invariant representations based on proposed agnostic objectives with the augmented data. A zero-shot adapter is utilized to help the learned global model bridge a domain gap between seen and unseen clients.

**Summary Of The Review:**

Overall, if authors can address my concerns of the generalization ability of client-invariant features and the experiment settings, I recommend to accept this paper.  The studied problem is important, and few current works have studied it. And the proposed method shows some novelty.

---

> ### Author Response · Authors · 2022-11-14
> **Reply to Reviewer LXCP (2/2)**
>
> **Q3: There is no theoretical analysis of the generalization performance.**
>
> **A:** We didn't show theoretical analysis of the generalization performance, and this is a limitation of our work. Normalization-based domain generalization methods [4, 5, 6] exploited the properties of instance and batch norm statistics for improving the generalization performance. Following these studies, we can intuitively expect that the model learns client-invariant representations on both original and augmented features with mixed instance and global statistics, and these representations generalize well on unseen domains. In addition, we empirically showed the generalization ability through extensive experiments on multiple DG benchmarks.
>
> [4] Hyeonseob Nam and Hyo-Eun Kim. Batch-instance normalization for adaptively style-invariant neural networks. Advances in Neural Information Processing Systems (NeurIPS), 2018.\
> [5] Seokeon Choi and Taekyung Kim and Minki Jeong and Hyoungseob Park and Changick Kim. Meta batch-instance normalization for generalizable person re-identification. Proceedings of the IEEE/CVF conference on Computer Vision and Pattern Recognition (CVPR), 2021.\
> [6] Kaiyang Zhou, Yongxin Yang, Yu Qiao, and Tao Xiang. Domain generalization with mixstyle. In International Conference on Learning Representations (ICLR), 2021.

---

> ### Author Response · Authors · 2022-11-14
> **Reply to Reviewer LXCP (1/2)**
>
> Dear reviewer,
>
> Thank you for your review and thoughtful comments. We appreciate you finding our paper well-motivated and well-written. We clarify your concerns regarding generalizability to unseen clients (domains) and expandability to more realistic scenario in cross-silo FL.
>
> We first emphasize that not only MixIG for client-invariant features but also the proposed zero-shot adapter has high novelty. It is the first work to adaptively estimate the interpolation parameter between test instance-level and global feature statistics as an instance-wise manner for inference.
>
> **Q1: How to ensure the claimed client-invariant features can generalize well on un-seen clients.**
>
> **A:** In multiple source domain generalization, the model can learn domain-invariant representations by finding common semantic features of each category existing in multiple datasets without any explicit consideration of unseen domains [1, 2]. Even though global statistics only include training clients (not include unseen clients), the model can extract client-invariant features by allowing local clients to build common semantic features across all clients in line with the multiple source domain generalization.
> It is noting that, for improving generalizability to unseen clients further, we utilize instance statistics of a test input by bridging a domain gap between training and test distribution with the proposed zero-shot adapter.
>
> [1] Ishaan Gulrajani and David Lopez-Paz. In search of lost domain generalization. In International Conference on Learning Representations (ICLR), 2021.\
> [2] Seonguk Seo, Yumin Suh, Dongwan Kim, Geeho Kim, Jongwoo Han, and Bohyung Han. Learning to optimize domain specific normalization for domain generalization. In European Conference on Computer Vision (ECCV), 2020.
>
> **Q2: The experiment setting is limited, e.g., only four clients are considered. The cross-silo FL may have much more clients.**
>
> **A:** Thank you for pointing out our experiment setup. In this paper, we followed the experimental setting of federated DG in [3], which has the small number of clients. However, we also think that current setup is very limited because the number of clients is small, non-IID label distribution (label skew) is not considered, and local iterations are long enough.
>
> We conducted two additional experiments following the reviewer’s comments. We first expand 3 training clients to 30 clients by distributing each dataset to 10 clients, which makes the client have the small amount of data from one domain. For IID label distribution, the data is equally distributed to 10 clients. For non-IID label distribution, each client has different class distribution across clients, and data split for this setup is formed from Dirichlet distribution ($\alpha = 0.5$). We illustrated class distribution of clients in Appendix A.6. In this setup, we randomly select 10 clients for local training with 20 iterations on each round, and total 120 rounds are conducted. We compared FedProx, FedBN, and our FedIG-A as belows:
>
> Table 1: Performance (%) on IID label distribution with 30 training clients.
> |Method|P|A|C|S|Avg|
> |------|---|---|---|---|---|
> |FedProx|94.95 (0.42) | 73.97 (0.48) | 70.72 (0.28) | 73.76 (0.24) | 78.35|
> |FedBN|94.53 (0.85) | 75.95 (0.17) | 70.53 (2.03) | 78.45 (2.16) | 79.86|
> |FedIG-A (ours)|94.76 (1.14) | 84.20 (0.31) | 79.20 (2.02) | 82.87 (1.76) | 85.26|
>
> Table 2: Performance (%) on non-IID label distribution with 30 training clients.
> |Method|P|A|C|S|Avg|
> |------|---|---|---|---|---|
> |FedProx|92.25 (0.25) | 71.75 (3.56) | 74.87 (0.61) | 69.30 (6.14) | 77.04|
> |FedBN|92.82 (0.13) | 74.78 (1.55) | 75.50 (0.95) | 71.71 (2.66) | 78.70|
> |FedIG-A (ours)|93.62 (0.30) | 79.54 (0.83) | 82.02 (0.15) | 77.49 (1.17) | 83.17|
>
> Our method significantly boosts the performance even when the number of clients is large. The performance is slightly downgraded in non-IID label distribution, but we can expect that this negative effect is alleviated if the method for non-IID label distribution is added together. We added the experimental details and results on Appendix A.6.
>
> [3] Junkun Yuan, Xu Ma, Defang Chen, Kun Kuang, Fei Wu, and Lanfen Lin. Collaborative semantic aggregation and calibration for separated domain generalization. arXiv e-prints, 2021.

---

> > ### Comment · Reviewer_LXCP · 2022-11-15
> > **Thanks for the response**
> >
> > Thanks for the response and the further experimental results from authors.
> >
> > Experiments of training clients to 30 clients by distributing each dataset to 10 clients well illustrates the scalability of the proposed method. However, the non-IID degree is relatively low. Many classic and popular FL works explore the $a=0.1$ in Dirichlet distribution to simulate the Non-IID data. Could your further conduct experiments with this higher Non-IID degree to verify the performance of the method?
> >
> > [1] Tackling the objective inconsistency problem in heterogeneous federated optimization. Wang et al. In NeurIPS 2020.
> >
> > [2] Adaptive Federated Optimization. Reddi et al. In ICLR 2021.
> >
> > [3] Federated Learning on Non-IID Data Silos: An Experimental Study. Li et al. In ICDE 2022.

---

> > > ### Author Response · Authors · 2022-11-16
> > > **Reply to Reviewer LXCP**
> > >
> > > Thanks for thoughtful suggestion.
> > >
> > > We conducted experiments with non-IID label distribution using Dirichlet distribution with $\alpha=0.1$ following many classic and popular FL works. We illustrated class distribution in Appendix A.6.
> > >
> > > Table 3: Performance (%) on non-IID label distribution ($\alpha=0.1$) with 30 training clients.
> > >
> > > |Method | P | A | C | S | Avg |
> > > |-|-|-|-|-|-|
> > > |FedProx | 83.17 (0.42)| 62.65 (4.42)| 66.95 (0.95)| 55.34 (4.33)| 67.03|
> > > |FedBN | 84.41 (1.40)| 61.62 (0.48)| 69.05 (3.81)| 59.85 (4.00)| 68.73|
> > > |FedIG-A (ours) | 92.13 (0.89)| 73.29 (2.90) |73.19 (1.12)| 71.96 (0.34)| 77.64|
> > >
> > > FedIG-A significantly improves the performance compared to the baselines. It demonstrates that FedIG-A can effectively deal with domain shift problems even in the case of severe non-IID label distribution.

---

> > > > ### Comment · Reviewer_LXCP · 2022-11-16
> > > > **Thanks for the more experiments**
> > > >
> > > > The results shows the FedIG-A could provide higher performance improvements when $a=0.1$  than $a=0.5$. Thus, this method shows its potential to tackle the heterogeneous data distribution. I'd like to increase score.

---

> > > > > ### Author Response · Authors · 2022-11-16
> > > > > **Reply to Reviewer LXCP**
> > > > >
> > > > > We are very grateful to the reviewer for raising the score.
> > > > >
> > > > > With the reviewer's comments, we could further show the effectiveness of our method and its potential to solve both domain and label shifts across clients.

---

### Official Review · Reviewer_p4rT · 2022-10-22

**Confidence:** 4
**Correctness:** 4
**Technical Novelty And Significance:** 3
**Empirical Novelty And Significance:** 3
**Recommendation:** 6

**Clarity, Quality, Novelty And Reproducibility:**

Generally speaking, this work is written with good clarity and quality. The novelty is fair and the originality is good.

**Strength And Weaknesses:**

Strength:

-- The designed two strategies are interesting and technically correct (especially the zero-shot adaptor). Most prior works conduct linear interpolation based on a pre-defined and fixed parameter, which is believed sub-optimal. While in this work, the authors designed an adaptive parameter estimation method to estimate the parameter.
-- The writing is clear and easy to follow. I feel no difficulty understanding the technical details of this work. The experimental results are extensive.

Weakness:

-- The work is heavily dependent on FedBN. The main difference is that author of this work designed an adaptive interpolation parameter estimation method. This jeopardizes the novelty and technical contribution of the whole work.
-- I am a little conservative about Eq. 4. If Eq. 4 stands, does that mean the u^l in Eq.3 tends to be 1?
-- The improvement of the designed solutions in Table 5, is not significant on some datasets. For example, on OfficeHome, the CSAC achieves 64.35, and the proposed solution achieves 64.71, which is a marginal improvement.

**Summary Of The Paper:**

This work targets federated domain generalization.  The solution is based on FedBN, which disentangles local model parameters into two groups: domain-specific and domain-invariant. Specifically, the authors designed two strategies: (1) client-agnostic learning with mixed instance-global statistics on BN parameters, as well as designed learning loss terms; (2) a zero-shot adaptor to estimate the interpolation parameter.

**Summary Of The Review:**

The ideas and designed solutions are interesting and technically correct. The experiments are extensive. The writing is clear.
What concerns me the most is the technical contribution of this work as it is heavily dependent on FedBN. Please provide more discussion and comparison to highlight your contribution in the rebuttal. I am willing to change my score if I can receive convincing feedback.

---

> ### Author Response · Authors · 2022-11-14
> **Reply to Reviewer p4rT (2/2)**
>
> **Q2: If Eq. 4 stands, does that mean the $u^l$ in Eq.3 tends to be $1$?**
>
> **A:** The motivation of MixIG is to augment local features with other clients’ statistics and learn client-invariant representations with the augmented features. With this motivation, we interpolate instance and global statistics with $u^l$ for generating augmented features. Here, $u^l$ is an interpolation weight vector, and its elements are sampled from uniform distribution $U(0,1)$ every iteration. Through random interpolation, we can generate more diverse features. We note that $u^l$ is not a learnable parameter, it is random weight vector for generating diverse domains randomly sampled at every iteration. About selecting $u^l$, we conducted ablation studies with fixed $u^l$ and randomly sampled $u^l$ in Table 1.
>
> In addition, the model is trained to build the same statistics with global statistics if we use the client-agnostic feature loss (Eq. 4) on every layer after BN layer and fix $u^l$ to $0$. However, we use the client-agnostic feature loss (Eq. 4) using the last features and use randomly generated $u^l$ for feature augmentations. Also, two client-agnostic loss functions are added to the main local cross-entropy loss function for helping learn client-invariant representations while minimizing cross-entropy loss on its own dataset.
>
> **Q3: The improvement of the designed solutions in Table 5, is not significant on some datasets.**
>
> **A:** In the case of OfficeHome, the domain shift problem is not as severe as other two benchmark datasets. Various papers [1, 2, 3] mentioned OfficeHome has less style variation across domains. We further analyze a domain gap across domains using the estimated statistics from the zero-shot adapter in Appendix A.4. With a slight domain gap across clients, the model can learn client-invariant representations without any domain generalization techniques, and the improvement of existing methods including our method is marginal. Similarly, in multiple source domain generalization, most methods marginally improve the performance compared to the baseline due to a small domain gap, e.g., the recent paper [4] improves the performance 64.72 to 65.89 compared to the baseline in OfficeHome. We note that our methods consistently improve the performance on three DG benchmarks regardless of the level of domain shifts within training clients or between training and test clients. If the domain gap is large, i.e., on PACS, our methods significantly outperform other methods. On VLCS and OfficeHome, our FedIG-A gets the SOTA performance or competitive performance with the SOTA method.
>
> [1] Kaiyang Zhou, Yongxin Yang, Timothy Hospedales, and Tao Xiang. Learning to Generate Novel Domains for Domain Generalization, European Conference on Computer Vision (ECCV), 2020.\
> [2] Dapeng Du, Jiawei Chen, Yuexiang Li, Kai Ma, Gangshan Wu, Yefeng Zheng, and Limin Wang.
> Cross-domain gated learning for domain generalization. International Journal of Computer Vision (IJCV), 2022.\
> [3] Guile Wu and Shaogang Gong. Collaborative Optimization and Aggregation for Decentralized Domain Generalization and Adaptation, In Proceedings of the IEEE/CVF International Conference on Computer Vision (ICCV), 2021.\
> [4] Juwon Kang, Sohyun Lee, Namyup Kim, and Suha Kwak. Style Neophile: Constantly Seeking Novel Styles for Domain Generalization. In Proceedings of the IEEE/CVF Conference on Computer Vision and Pattern Recognition (CVPR), 2022.

---

> ### Author Response · Authors · 2022-11-14
> **Reply to Reviewer p4rT (1/2)**
>
> Dear reviewer,
>
> Thank you for your review and valuable comments. We appreciate your interest for our ideas and designed solutions. We clarify your concerns regarding dependency of proposed methods to FedBN.
>
> We first emphasize that not only the zero-shot adapter but also our MixIG for privacy-preserving multi-source domain generalization is the first work to learn domain-invariant representations with other domains’ statistics while keeping data private.
>
> **Q1: The work is heavily dependent on FedBN. The main difference is that author of this work designed an adaptive interpolation parameter estimation method.**
>
> **A:** We first describe why we start with FedBN, then we discuss about the dependency of proposed methods to FedBN.
>
> Why we start with FedBN:\
> FedBN is proposed to build the global model when domain shift exists across clients (see the performance gap between FedAvg and FedBN in Table 5). In FedBN, BN layers remain local, i.e., a client-specific part, and the remaining parts are uploaded to the server, i.e., a client-agnostic part. FedBN does not modify the architecture itself, only initialization process is changed. At each round, the client-specific part keeps its parameters, and the client-agnostic part is initialized with the server model’s parameters. Since this simple modification allows to build the server model even when domain shift exists across clients, we start with FedBN as the baseline. However, our proposed methods can be applied to naive framework, e.g., FedAvg, as described below.
>
> The dependency of each method to FedBN:\
> In this paper, we propose two strategies: (1) MixIG for learning client-invariant representations in local training; (2) the zero-shot adapter for bridging a domain gap between training and test clients at inference time.
>
> (1) The motivation of MixIG is to augment local features with randomly interpolated instance and global statistics from the server, while keeping the privacy of the source domains. The concept of our feature augmentation from instance and global statistics is equally applicable to any framework that has BN layers. We conducted experiments using MixIG and client-agnostic learning (CAL) on FedAvg. Here, the only difference between FedAvg and FedBN is $\mu_k$ and $\sigma_k$ in Fig. 2, where FedAvg initializes $\mu_k$ and $\sigma_k$ with $\mu_G$ and $\sigma_G$ every round but FedBN keeps $\mu_k$ and $\sigma_k$.
>
> Table 1: Performance (%) of FedAvg with our MixIG.
> |Method|P|A|C|S|Avg|
> |-|-|-|-|-|-|
> |FedAvg|90.40 (0.97)|72.52 (2.28)|72.59 (0.37)|73.90 (1.74)|77.35|
> |FedAvg w/ MixIG (ours)|93.11 (0.08)|80.05 (2.52)|74.00 (0.09)|79.40 (0.41)|81.69|
> |FedBN|92.74 (0.97)|77.08 (0.59)|75.08 (0.85)|78.28 (1.14)|80.79|
> |FedBN w/ MixIG (ours)|92.99 (0.61)|82.17 (1.15)|77.71 (1.13)|83.40 (0.19)|84.07|
>
> Note that the performance in the last two rows is reported in Table 3. Our MixIG boosts the performance of FedAvg as well as FedBN. Regardless of the baseline methods, our client-agnostic learning with MixIG helps the model learn client-invariant representations.
>
> (2) Our zero-shot adapter is totally not dependent on FedBN. At inference time, the zero-shot adapter estimates the parameter for linear interpolation based on the difference between instance and global statistics. We conducted experiments to show the effectiveness of the zero-shot adapter even when we use FedAvg.
>
> Table 2: Performance (%) of FedAvg with our zero-shot adapter.
> |Method|P|A|C|S|Avg|
> |-|-|-|-|-|-|
> |FedAvg|90.40 (0.97)|72.52 (2.28)|72.59 (0.37)|73.90 (1.74)|77.35|
> |FedAvg-A (ours)|93.11 (0.08)|80.93 (0.79)|77.01 (0.24)|82.54 (0.79)|83.40|
>
> Here, FedAvg-A indicates FedAvg with the zero-shot adapter. Interestingly, FedAvg with the zero-shot adapter significantly improves the performance. FedAvg and FedAvg-A have the same feature extractor and classifier (both architecture and parameters). The zero-shot adapter learns how to interpolate instance and global statistics through federated learning, and we can see that the trained adapter is effectively working on unseen domains. It aims to improve the generalizability of the model.
>
> Finally, we applied our two methods together to FedAvg, and the results are as below:
>
> |Method|P|A|C|S|Avg|
> |-|-|-|-|-|-|
> |FedIG-A based on FedAvg (ours)|93.05 (0.25)|82.30 (1.42)|77.60 (0.72)|83.94 (0.68)|84.22|
> |FedIG-A based on FedBN (ours)|94.24 (0.33)|84.30 (0.44)|79.80 (1.16)|83.79 (0.49)|85.53|
>
> FedIG-A based on FedAvg also achieves the state-of-the-art performance compared to other competitive methods. This result also shows that our two proposed methods complement each other.
>
> Furthermore, the concept of the zero-shot adapter is to adaptively estimate the proper statistics as an instance-wise manner, so we expect that our zero-shot adapter can be utilized in other tasks where training and test distributions are different, e.g., test-time adaptation and multi-source domain generalization. Our future work is to expand our work to other tasks.

---

### Official Review · Reviewer_RmwN · 2022-10-22

**Confidence:** 3
**Correctness:** 3
**Technical Novelty And Significance:** 2
**Empirical Novelty And Significance:** 3
**Recommendation:** 5

**Clarity, Quality, Novelty And Reproducibility:**

The presentation of the paper should be improved. The idea is interesting, which is mainly built on FedBN (Li et al., 2021b). I think the authors sufficiently explained the experimental setup.

**Strength And Weaknesses:**

Strength:

The main strength of the paper is the experimental results, which are provided in an organized manner covering a broad range or DG benchmarks and comparing with a sufficiently number of baselines. However, the reviewer has major concerns regarding the problem itself, the proposed method, and some claims in the paper:


Weaknesses:

This paper addresses adaptation to a new completely unseen test domain. The main goal of FL is to help clients who participate in training in terms of obtaining a global model, which addresses their own test error on their own test distributions. Otherwise, why should a client participate in training, which is aimed to optimize a model for a completely different test domain?

The introduction of zero-shot adapter for example increases computational costs for local clients while it does not directly benefit their own test error. I am not convinced that a client is willing to participate in such collaborative learning. For example, consider cross-silo FL where multiple hospitals plan to learn a model collaboratively. Their main motivation is to obtain a model that helps their own patients at the inference time.

-----

Another major issue is that what if the model used to train does not have batch normalization layers. It may be a simple CNN. In particular, batch nomlization has been shown to suffer from 1) increasing  computational primitive, which incurs memory overhead 2) introducing hidden hyper-parameters that have to be tuned 3) breaking the independence between training examples in the minibatch (Brock ICLR21 and ICML21).

Andy Brock, Soham De, Samuel L. Smith, and Karen Simonyan. "High-performance large-scale image recognition without normalization." In International Conference on Machine Learning (ICML) 2021.

Andy Brock, Soham De, and Samuel L. Smith. "Characterizing signal propagation to close the performance gap in unnormalized ResNets." In International Conference on Learning Representations (ICLR) 2021.

-----

The authors assumed that client $k$ has a single domain data. What if client $k$ has different domains across training/test time, which is a natural scenario.

-----

"local models are easily over-fitted to their domains"

This happens only when a model is trained locally for several iterations. If the model is aggregated frequently, this will not happen in standard FL training.

The authors proposed to locally train $\{\theta_k^a,\theta_k^s,\phi_k\}$ for long epochs. There are several work in distributed learning, which show that local updates are not quite useful both experimentally and theoretically (Karimireddy ICML20). It will be nice if the authors explain why local training for long epochs is useful in this problem?

Sai Praneeth Karimireddy, Satyen Kale, Mehryar Mohri, Sashank Reddi, Sebastian Stich, and Ananda Theertha Suresh. "Scaffold: Stochastic controlled averaging for federated learning." In International Conference on Machine Learning (ICML) 2020.

-----

For MixIG: clients still need to share BN statistics to be averaged so the claim that this method has the same privacy guarantees as FedAvg is not accurate. Indeed, in addition to sharing $\theta_k,\phi_k$, sharing those statistics leads to additional privacy leakage.

-----

The interpolation mean vector is drawn uniformly at random  from $(0,1)$. Why did the authors choose this specific distribution? It seems to me that these weights should be considered as additional hyperparameters. Another major issue is that whether those weights are client-specific or shared among clients? Based on current notation, it seems that all clients use the same weights for BN layer $l$.

This paper has two additional hyperparameters $\lambda_1$ and $\lambda_2$., which makes the solution less practical given the challenges of tuning these parameters.

-----

Minor comments:

Typo in Section I: "Af inference time"

The first phrase in the second sentence of paragraph "Challenges:" is incomplete.

Typo in the first sentence of paragraph: "Mixing Instance and Global Statistics (MixIG):"

**Summary Of The Paper:**

This paper studies how to learn a global model in FL from multiple distributed source domains and generalize the model to new clients in unseen domains at inference time. The authors propose client-agnostic learning with mixed instance-global statistics
for local training along with  zero-shot adaptation with estimated statistics for inference, which can be viewed as a privacy-preserving multi-source domain generlaization method. To obtain client-invariant representations, the authors augment local features using data distribution of other clients via aggregated BN statistics from the global model.



**Summary Of The Review:**

While the paper has merits mainly w.r.t. experimental results, there are major concerns regarding the problem itself, the proposed method, and some claims in the paper

---

> ### Author Response · Authors · 2022-11-14
> **Reply to Reviewer RmwN (4/4)**
>
> **Q6: For MixIG: clients still need to share BN statistics to be averaged so the claim that this method has the same privacy guarantees as FedAvg is not accurate. Indeed, in addition to sharing $\theta_k$, $\phi_k$, sharing those statistics leads to additional privacy leakage.**
>
> **A:** We first emphasize that local clients do not share their BN statistics with other clients. Local clients transmit all parameters to the server, then only the aggregated parameters (including global statistics) are shared to clients. It is the same process with FedAvg.
>
> The communication process of FedIG-A is as follows: after local training, all parameters including local BN statistics are uploaded to the server, and all parameters are aggregated in the server. Then, the aggregated parameters are broadcasted to each client. The process is exactly same with FedAvg, where clients upload all parameters including its BN statistics to the server and receive the aggregated parameters. The only difference between FedIG-A and FedAvg is that the client in FedIG-A has two versions of statistics, local BN statistics and global BN statistics. In MixIG, we use instance and global statistics, not other domains’ BN statistics. As the reviewer mentioned, it causes severe privacy leakage if we share client-specific parts across other clients, but our FL does not share client-specific parts.
>
> **Q7: Why did the authors choose this specific distribution for the interpolation mean vector. Another major issue is that whether those weights are client-specific or shared among clients? Based on current notation, it seems that all clients use the same weights for BN layer l.**
>
> **A:** We note that $u$ in all BN layers is randomly sampled from the uniform distribution at every iteration, and we choose uniform distribution to equally mix instance and global statistics. As the reviewer mentioned, distribution for generating $u^l$ may be considered as an addition hyperparameter. We conducted experiments using uniform and gaussian distribution with different range.
>
> Table 5: Performance (%) of FedIG-A with different distribution for $u^l$.
> |Method|P|A|C|S|Avg|
> |-|-|-|-|-|-|
> |N(0.5, 0.1) | 94.70 (0.38) | 83.52 (0.93) | 79.97 (0.09) | 83.42 (1.03) | 85.40|
> |N(0, 0.1) | 93.41 (0.33) | 76.27 (0.83) | 74.49 (1.02) | 80.43 (0.09) | 81.15|
> |U(0, 1) | 94.24 (0.33) | 84.30 (0.44) | 79.80 (1.16) | 83.79 (0.49) | 85.53|
> |U(0, 0.5) | 94.46 (0.21) | 84.50 (0.17) | 80.97 (1.15) | 84.46 (0.09) | 86.10|
> |U(0.5, 1) | 94.58 (0.21) | 82.89 (0.59) | 78.14 (1.48) | 83.18 (0.04) | 84.69|
> |U(-0.1, 1.1) | 94.46 (0.64) | 83.64 (0.83) | 80.12 (0.36) | 84.59 (0.05) | 85.70|
>
> As shown in the above table, the performance is similar regardless of the distribution type, but the range of $u^l$ affects the performance. In addition, if $u^l$ has negative values, instance and global statistics are extrapolated. MixIG with extrapolation generates more diverse data from unseen domains, e.g., U(-0.1, 1.1), but there is a risk that strange data may be generated, e.g., the performance is severely degraded when using N(0, 0.1). Although the distribution type and the range are different across domains for the best performance, we set the distribution to U(0,1). The performance is not sensitive to the distribution type, and the range of [0,1] is reasonable.
>
> **Q8: This paper has two additional hyperparameters $\lambda_1$ and $\lambda_2$, which makes the solution less practical given the challenges of tuning these parameters.**
>
> **A:** We conducted ablation studies about hyperparameters $\lambda_1$ and $\lambda_2$ in Appendix A.1. Although the optimal hyperparameters are different with each domain, $\lambda_1$ in [0.1, 0.5] and $\lambda_2$ in [2.0, 5.0] show consistent results on all domains. We set $\lambda_1$ to 0.1 and $\lambda_2$ to 4.0 for all experiments.
>
> Note that we conducted additional experiments in more realistic settings, where the number of training clients is 30 and data is distributed with non-IID label distribution, i.e., label distribution skew, across clients (please see the experimental setup and the results in Appendix A.6).

---

> ### Author Response · Authors · 2022-11-14
> **Reply to Reviewer RmwN (3/4)**
>
> **Q4: The authors assumed that client k has a single domain data. What if client k has different domains across training/test time, which is a natural scenario.**
>
> **A:** As the reviewer mentioned, it is a natural scenario that different domain data exists in one client. There are two practical scenarios: (1) multiple domains in each client during training; (2) multiple domains in the client at the test time.
>
> (1) Our concept of MixIG is to learn client-invariant representations using other clients' characteristics. For this goal, we augment local features with global statistics to indirectly access distribution of other clients. Regardless of the number of domains in the client, local statistics reflect local domains and global statistics reflect all clients’ domains. Therefore, our approach can be simply applied to learn client-invariant representations even when the client has various domains. We conducted experiments when multiple domains exist on each client during training. Here, we use 3 clients, and each client has two domains, e.g., 3 clients have P and A, A and C, C and P domains, respectively, when testing on S domain.
>
> Table 3: Performance (%) on the scenario where each client has multiple domains.
> |Method|P|A|C|S|Avg|
> |-|-|-|-|-|-|
> |FedProx | 95.84 (0.13) | 81.13 (2.24) | 75.98 (0.48) | 74.51 (2.32) | 81.86|
> |FedBN | 95.72 (0.13) | 80.91 (0.14) | 75.96 (0.27) | 75.82 (0.83) | 82.10|
> |FedIG-A (ours) | 97.82 (0.30) | 85.11 (0.00) | 78.43 (1.30) | 82.12 (0.67) | 85.35|
>
> If the local client has multiple domain data, the local model can learn domain-invariant representations without any domain generalization algorithms. FedProx achieves high performance compared to the previous setting, where each client has single domain data. Our method shows similar performance with the pervious setting. In addition, we can use multiple local BN layers for each client to learn client-specific characteristics, separately for multiple domains. Then, we can more effectively apply our algorithms to this scenario.
>
> (2) Our approach is directly applied to the second case, i.e., multiple domains at the test time. At inference time, we use the model with the linear interpolated BN statistics between instance and global statistics, and the interpolation parameter is generated by the zero-shot adapter that works as an instance-wise manner. Our proposed inference method, i.e., the zero-shot adapter, depends on the input not domains, so it equally works on multiple domains. It does not have any assumptions about the number of domains, thus our method can work independently on the number of domains in seen or unseen clients.
>
> **Q5: "local models are easily over-fitted to their domains" This happens only when a model is trained locally for several iterations. If the model is aggregated frequently, this will not happen in standard FL training.**
>
> **A:** In federated DG, each client trains the local model for long epochs to reduce communication costs following [8]. However, as the reviewer mentioned, the model can be aggregated frequently with short local epochs in situations where frequent communication is possible. We conducted experiments with short local iterations and long total rounds (10 times shorter local iterations and 10 times longer total rounds compared to the previous setting).
>
> Table 4: Performance (%) on FL models with short local iterations.
> |Method|P|A|C|S|Avg|
> |-|-|-|-|-|-|
> |FedProx | 92.40 (0.47) | 74.93 (0.03) | 72.25 (0.03) | 72.28 (0.22) | 77.88|
> |FedBN | 93.09 (0.34) | 75.42 (0.17) | 73.48 (0.61) | 76.97 (2.21) | 79.74|
> |FedIG-A (ours) | 93.56 (0.47) | 84.89 (0.24) | 79.91 (0.60) | 83.98 (0.49) | 85.58|
>
> We get the similar results with short local iterations. Even with small local iterations, the model is fitted to each domain, which makes it difficult to build a generalized global model. Additionally, our proposed methods work well independent of the length of local iterations, as shown in the above table.
>
> [8] Junkun Yuan, Xu Ma, Defang Chen, Kun Kuang, Fei Wu, and Lanfen Lin. Collaborative semantic aggregation and calibration for separated domain generalization. arXiv e-prints, 2021.

---

> ### Author Response · Authors · 2022-11-14
> **Reply to Reviewer RmwN (2/4)**
>
> **Q3: Another major issue is that what if the model used to train does not have batch normalization layers. It may be a simple CNN. In particular, batch normalization has been shown to suffer from 1) increasing computational primitive, which incurs memory overhead 2) introducing hidden hyper-parameters that have to be tuned 3) breaking the independence between training examples in the minibatch (Brock ICLR21 and ICML21).**
>
> **A:** Most widely used CNNs exploit batch normalization (BN) layers [1, 2], and many recent methods [3, 4, 5] have been consistently proposed on batch normalization layers. Our proposed method, FedIG-A, utilizes the characteristics of BN layers in line with the previous BN-based architectures and methods.
> As the reviewer mentioned, some works pointed out the disadvantages of BN layers and designed the new architecture without using BN layers. In this case, we cannot directly apply our methods, and this is our limitation. However, we can expect that our work is expanded to the model not having BN layer. Our concept of client-agnostic learning is to utilize both local and global characteristics when training the local models, and we exploit BN layers because they contain these characteristics. Even though the model does not have BN layers, some parts of the model contain domain-specific information, e.g., parts of the model [6] or hyper networks [7] in personalized FL. If we interpolate these parts between local and global models, we expect that our client-agnostic learning works well though we haven’t tried any other experiments.
>
> [1] Kaiming He and Xiangyu Zhang and Shaoqing Ren and Jian Sun. Deep residual learning for image recognition. In Proceedings of the IEEE conference on computer vision and pattern recognition (CVPR), 2016.\
> [2] Mingxing Tan, and Quoc Le. Efficientnet: Rethinking model scaling for convolutional neural networks. International conference on machine learning (ICML), 2019.\
> [3] Chaoqi Chen, Jiongcheng Li, Xiaoguang Han, Xiaoqing Liu, and Yizhou Yu. Compound domain generalization via meta-knowledge encoding. In Proceedings of the IEEE Conference on Computer Vision and Pattern Recognition (CVPR), 2022.\
> [4] Ziqi Zhou, Lei Qi, Xin Yang, Dong Ni, and Yinghuan Shi. Generalizable cross-modality medical image segmentation via style augmentation and dual normalization. In Proceedings of the IEEE Conference on Computer Vision and Pattern Recognition (CVPR), 2022.\
> [5] Taesik Gong, Jongheon Jeong, Taewon Kim, Yewon Kim, Jinwoo Shin, and Sung-Ju Lee. Robust continual test-time adaptation: Instance-aware bn and prediction-balanced memory. Advances in Neural Information Processing Systems (NeurIPS), 2022.\
> [6] Benyuan Sun and Hongxing Huo and Yi Yang and Bo Bai. Partialfed: Cross-domain personalized federated learning via partial initialization. Advances in Neural Information Processing Systems (NeurIPS), 2021.\
> [7] Aviv Shamsian and Aviv Navon and Ethan Fetaya and Gal Chechik. Personalized federated learning using hypernetworks. International Conference on Machine Learning (ICML), 2021.

---

> ### Author Response · Authors · 2022-11-14
> **Reply to Reviewer RmwN (1/4)**
>
> Dear reviewer,
>
> Thanks for your review and thorough comments. We appreciate that you found our paper well organized with rich experiments. We clarify your concerns regarding the problem itself, the proposed method, and our claims.
>
> **Q1: The main goal of FL is to help clients who participate in training in terms of obtaining a global model, which addresses their own test error on their own test distributions. Why should a client participate in training, which is aimed to optimize a model for a completely different test domain?**
>
> **A:** Our target task, federated domain generalization, aims to solve two challenges in federated learning: (1) the difficulty of building the global model with local training clients from different domains; (2) low generalizability to test clients, where data distribution deviates from those of training clients. It is a realistic scenario where clients have different domains with other clients, and the solution for the first challenge aims to obtain a global model to address test error on their own test distributions in line with the main goal of FL as the reviewer mentioned.
>
> We showed the performance of FedAvg and FedIG-A on seen domains (inside the federation).
>
> Table 1: Performance (%) on seen domains.
> |Method|P|A|C|S|Avg|
> |-|-|-|-|-|-|
> |FedAvg | 95.60 (1.91) | 87.29 (3.91) | 90.82 (1.62) | 92.02 (1.09) | 91.43|
> |FedIG-A (ours) | 97.40 (0.79) | 93.53 (1.77) | 94.53 (1.08) | 95.33 (0.44) | 95.20|
>
> In many practical cases, local clients have different domain data, and it hinders building the global model as shown in the above results. Learning client-invariant representations (FedIG and FedIG-A) in local clients helps to build the global model (we showed the performance on participated clients in Appendix A.7) even though domain shift exists across clients.
>
> Moreover, in a realistic scenario, the distribution gap exists between training and test datasets even in the client inside the federation, and we can think that the ultimate goal of FL is to improve the performance on both seen and unseen domains. We report the performance on unseen domains as below:
>
> Table 2: Performance (%) on unseen domains.
> |Method|P|A|C|S|Avg|
> |-|-|-|-|-|-|
> |FedAvg | 90.40 (0.97) | 72.52 (2.28) | 72.59 (0.37) | 73.90 (1.74) | 77.35|
> |FedIG-A (ours) | 94.24 (0.33) | 84.30 (0.44) | 79.80 (1.16) | 83.79 (0.49) | 85.53|
>
> Our methods treating federated DG shows good performance on both seen and unseen domains (seen and unseen clients) by learning client-invariant representations.
>
> **Q2: The introduction of zero-shot adapter for example increases computational costs for local clients while it does not directly benefit their own test error. I am not convinced that a client is willing to participate in such collaborative learning.**
>
> **A:** As the reviewer mentioned, using the zero-shot adapter increases the training/inference costs (Table 4), and the performance increase for seen domains is marginal (comparison between FedIG and FedIG-A in Table 12 in Appendix A.7). However, some participated clients have unseen domains when testing (test distribution shift). In this case, our zero-shot adapter helps the model dynamically adapt to each test instance even for the participated clients having unseen test domains. Given this advantage, clients might be willing to pay an extra cost for training the zero-shot adapter and using the zero-shot adapter for inference. Note that computational costs marginally increase as shown in Table 4. As we mentioned in the manuscript, users can choose to use the zero-shot adapter according to the requirements of the target devices.

---

> > ### Comment · Reviewer_RmwN · 2022-11-18
> > **Thanks for the response**
> >
> > I would like to thank the authors for their notes. Some comments are properly addressed for example regarding the privacy leakage. However, some major concerns are not addressed properly:
> >
> > The introduction of zero-shot adapter increases computational costs for local clients while it does not directly benefit their own test error. I am not convinced that a client is willing to participate in such collaborative learning.
> >
> > The authors responded "Note that computational costs marginally increase as shown in Table 4."
> >
> > I do not think $50$% increase in training time is marginal. It is also not known whether, e.g., FedBN can be trained for more iterations with additional tuning (we note that FedIG requires a number of additional hyperparameters) and achieves similar accuracies as shown in this paper.
> >
> > I appreciate that the authors acknowledged that they cannot directly apply their methods to simple CNNs, and this is their limitation. But this raises some concerns whether this method will be useful for cross-device FL where the memory of a device may not allow it to fit other types of models.

---

> > > ### Author Response · Authors · 2022-11-19
> > > **Reply to Reviewer RmwN (2/2)**
> > >
> > > **Q2: The authors responded "Note that computational costs marginally increase as shown in Table 4."I do not think 50% increase in training time is marginal. It is also not known whether, e.g., FedBN can be trained for more iterations with additional tuning (we note that FedIG requires a number of additional hyperparameters) and achieves similar accuracies as shown in this paper.**
> > >
> > > **A:** The increase of computational cost is small at inference time, but the increase is not marginal in training time, as the reviewer mentioned. We correct our reply as the increase of computational cost is not marginal in training. As advised by the reviewer, we conducted two experiments of FedBN with long local iterations (600 iterations) and long total rounds (120 rounds) that are three times longer than FedIG-A for a fair comparison in terms of training time. We note that our common hyper-parameters for all methods, e.g., learning rate, optimizer, batch size, and the number of local iterations and rounds, are from the previous work [8], not optimized to our FedIG-A. Although FedIG-A has several additional hyper-parameters, we showed FedIG-A is not highly dependent on hyper-parameters in Appendix. FedBN trains the local model with cross-entropy loss on local data, thus there is no hyper-parameter to be adjusted except the common hyper-parameters. We can tune the common hyper-parameters for FedBN, then the performance can be improved. However, the performance gap is not marginal between FedBN and FedIG-A, and we think that it might not be fully reduced by only tuning hyper-parameters.
> > >
> > > Table 8: Performance (%) of FedBN and FedIG-A with the same training time.
> > >
> > > |Method|P|A|C|S|Avg|
> > > |-|-|-|-|-|-|
> > > |FedBN w/ long local iterations|93.03 | 76.17 | 75.35 | 78.69|80.81 |
> > > |FedBN w/ long total rounds|93.09|76.76|76.65|78.43| 81.23|
> > > |FedIG-A (ours)| 94.24 (0.33) | 84.30 (0.44) | 79.80 (1.16) | 83.79 (0.49) | 85.53|
> > >
> > > Note that we report the results of the experiment with only one seed due to time limit. FedBN with long training achieves marginally improved performance compared to the previous setting (200 local iterations), but FedIG-A outperforms FedBN with a large gap regardless of the number of iterations or rounds. Due to domain shift across clients, local models cannot fully learn client-invariant representations, even being trained with long iterations or rounds.
> > >
> > > [8] Junkun Yuan, Xu Ma, Defang Chen, Kun Kuang, Fei Wu, and Lanfen Lin. Collaborative semantic aggregation and calibration for separated domain generalization. arXiv e-prints, 2021.
> > >
> > > **Q3: I appreciate that the authors acknowledged that they cannot directly apply their methods to simple CNNs, and this is their limitation. But this raises some concerns whether this method will be useful for cross-device FL where the memory of a device may not allow it to fit other types of models.**
> > >
> > > **A:** For federated domain generalization, we first consider cross-silo FL. Our algorithms are applicable to the cross-silo FL setting because devices can use the same type of models having BN layers. We additionally showed the experimental results with 30 training clients having iid and non-iid label distribution in [our reply](https://openreview.net/forum?id=S4PGxCIbznF&noteId=gUgdZ-nWk2) following the comments from Reviewer LXCP. Our method cannot be directly applied in cross-device FL, where clients might have different architectures. We leave federated domain generalization in cross-device FL to future work.

---

> > > ### Author Response · Authors · 2022-11-19
> > > **Reply to Reviewer RmwN (1/2)**
> > >
> > > Dear reviewer,
> > >
> > > Thanks for your thoughtful comments. We hope our additional responses address your concerns.
> > >
> > > **Q1: The introduction of zero-shot adapter increases computational costs for local clients while it does not directly benefit their own test error. I am not convinced that a client is willing to participate in such collaborative learning.**
> > >
> > > **A:** In a realistic scenario, the client inside the federation could have both in-domain and out-of-domain test data, where distribution of out-of-domain test data is deviated from training distribution. In this paper, the evaluation for unseen clients can be regarded as the evaluation for out-of-domains in the client participated in FL. Therefore, Table 3 in the paper demonstrated that our zero-shot adapter helps the client model to improve the performance of out-of-domain test data while boosting the performance of in-domain test data (please see Table 12 in Appendix). We reported the average performance on in-domain and out-of-domain data below.
> > >
> > > Table 6: Average performance (%) on in-domain and out-of-domain data.
> > > |Method|in-domain|out-of-domain|
> > > |-|-|-|
> > > |FedAvg|91.43|77.35|
> > > |FedIG (ours)|94.53|84.07|
> > > |FedIG-A (ours)|95.20|85.53|
> > >
> > > It shows that the adapter can directly benefit participated clients’ test error. Given this advantage, clients might pay an extra cost for training the zero-shot adapter.
> > >
> > > We emphasize that the zero-shot adapter is the first work to adaptively estimate the interpolation parameter between test instance-level and global feature statistics as an instance-wise manner for inference. We further show the effectiveness of the adapter with additional experiments. We conducted experiments only using our zero-shot adapter on FedAvg (without MixIG and FedBN).
> > >
> > > Table 7: Performance (%) of FedAvg with our zero-shot adapter.
> > > |Method|P|A|C|S|Avg|
> > > |-|-|-|-|-|-|
> > > |FedAvg|90.40 (0.97)|72.52 (2.28)|72.59 (0.37)|73.90 (1.74)|77.35|
> > > |FedAvg-A (ours)|93.11 (0.08)|80.93 (0.79)|77.01 (0.24)|82.54 (0.79)|83.40|
> > >
> > > Here, FedAvg-A indicates FedAvg with the zero-shot adapter. Interestingly, FedAvg with the zero-shot adapter significantly improves the performance. FedAvg and FedAvg-A have the same feature extractor and classifier (both architecture and parameters). The zero-shot adapter learns how to interpolate instance and global statistics through federated learning, and we can see that the trained adapter is effectively working on unseen domains. It shows that the adapter directly improves participated clients’ test error without any other algorithms.
> > >
> > > In [our reply](https://openreview.net/forum?id=S4PGxCIbznF&noteId=aEjGJWHTqHF) for Reviewer p4rT, we further showed the novelty of two proposed methods, separately. Please see our reply.

---

### Official Review · Reviewer_Rg7n · 2022-10-24

**Confidence:** 5
**Correctness:** 2
**Technical Novelty And Significance:** 2
**Empirical Novelty And Significance:** 2
**Recommendation:** 3

**Clarity, Quality, Novelty And Reproducibility:**

The writing is not good enough. Though I can understand the main idea of the paper, the details in model design are hard to understand. Based on current presentation, the reproducibility will also be very difficult.

**Strength And Weaknesses:**

Strength:

* The experiments are comprehensive and results are good.
* The method is simple and lightweight.

Weaknesses:

First of all, I would say that the paper is not well written. Though the method is simple, it is not easy to understand all the components as a unified model. Below I give my comments as well as the parts that are not clear to me.

* My major concern involves with the core model design. It is not reasonable for me that the method aims to keep two groups of BN parameters at each BN layer, at the same time to let the features generated by the two parameter groups be same by Eq.4. Would this make the BN parameters same? I am wondering how will a simplified model performs, e.g., by dropping all local BN, $\mathcal{L}_{CE}$ in Eq.6 and the loss in Eq.4.
* It is not clear to me how the  output features of the two BN layers are fused? Of if they are not fused, the model will cause the increase of computational cost in local clients and some analysis should be provided.
* From a unified perspective, I am wondering is $\alpha$ in Eq.7 equivalent to $\mu$ in Eq.3, only that $\alpha$ are dynamically generated rather than sampling $\mu$ from a gaussian distribution. If it is this case, I would suggest to use the same symbol to ease the understanding.
* The paper presents that "a zero-shot adapter that is carefully designed to dynamically generate $\alpha$ for each input in both seen and unseen domains". Then I have two questions:
    * The $L_A$ in Eq.8 requires label $y_{i,k}$ for loss computation. However, it is not clear where the labels are obtained for unseen domains since in Sec. 4.1, it mentions that the unseen domain is only used for evaluation, which means that the data are not labeled.
    * Regarding the specific implementation in Eq.9, a MLP is used for generating $\delta$ and $\epsilon$. I am concerning whether the network design is promising. Is it necessary to constraint the range of the two values so as to avoid generating unexpected values (e.g., 10000).
* The statement "we use diverse augmented domain from other clients beyond regularization." is not clear.

**Summary Of The Paper:**

This article tackles the problem of federated domain generalization. The method includes two parts: the first part mixes the BN parameters of each local model with the global model, and performs feature augmentation to improve model robustness; the second part introduces a zero-shot adaptor for unseen client generation. The method is verified over three datasets and results look promising.

**Summary Of The Review:**

The paper needs a significant revision to improve the presentation and clarify the rational of the model design. The current version is far from reaching the acceptance bar of ICLR.

---

> ### Author Response · Authors · 2022-11-15
> **Reply to Reviewer Rg7n (4/4)**
>
> **Q4: The $L_A$ in Eq.8 requires label $y_{i,k}$ for loss computation. However, it is not clear where the labels are obtained for unseen domains since in Sec. 4.1, it mentions that the unseen domain is only used for evaluation, which means that the data are not labeled.**
>
> **A:** Sorry for confusing the training and inference stages for the zero-shot adapter. As the reviewer mentioned, the zero-shot adapter is designed to estimate the proper statistics for each test input at the test time. However, randomly initialized zero-shot adapter cannot work well as shown in Table 3 (Random alpha). In local clients, we train the zero-shot adapter with labeled source data using Eq. 8. Trained zero-shot adapter is then used for unseen domains (unseen clients) at the test time. We revised the paper about the zero-shot adapter for clear presentation.
>
> **Q5: Regarding the specific implementation in Eq.9, a MLP is used for generating $\delta$ and $\epsilon$. I am concerning whether the network design is promising. Is it necessary to constraint the range of the two values so as to avoid generating unexpected values (e.g., 10000).**
>
> **A:** We use the clamp function to $\alpha^l$ within 0 and 1. We revised the paper of the zero-shot adapter part. After federated learning, we plot the alpha values from all test samples in Appendix A.4. We can see that the zero-shot adapter is well trained with our network design and our training strategy.
>
> **Q6: The statement "we use diverse augmented domain from other clients beyond regularization." is not clear.**
>
> **A:** Here, our method generates more diverse domains' features using randomly mixed instance and global statistics and updates the model with the original and augmented features to learn client-invariant representations. Through this statement, we want to say that our method is not just a regularization method (like FedProx) which objective is not to deviate from the global parameters even though we exploit global statistics. For clear description, we revised it as “Different with the previous work (FedProx) that directly regularizes local weight parameters with the global model, our proposed learning considers the importance of weight parameters for client-invariant representations with diverse domain data.”
>
> We described experimental settings and implementation details in the paper for the reproducibility. Our first idea is to mix instance-level and global feature statistics, described in Eq. (3). It can be easily implemented by modifying BN layers in Pytorch. For improving the reproducibility of the zero-shot adapter, we revised the paper with more clear description. Please check the revised paper.

---

> > ### Comment · Reviewer_Rg7n · 2022-11-25
> > **post rebuttal comment**
> >
> > Thanks for the detailed response. I have read the response, other reviewers' comments, as well as the revised manuscript. But I still have concerns:
> >
> > * The first concern involves with Q1, a similar question is also raised by Reviewer p4rT (Q2). I understand that local BN and global BN play different roles in the network, however, the question is whether $\mu_i^l$ and $\mu_G^l$ will be equal under the constraint of Eq. 4.
> > * The second concern is about the new analysis provided in the context of Q5. Some statements in Appendix A.4 are not clear to me, e.g., "In S domain, alpha values at the low-level layers are lower than other domains. Since there is a large domain gap between S and other domains, distribution of alpha values are different." Is it reasonable that $\alpha$ should be learnt to be small for domain S which has a large gap to other domains? Or equivalently, should we expect to use global statistics $\mu_G$ (in Eq. 7) more here?
> > * The newly added statement "we set $\alpha^l$ as $\epsilon^l$, which is the mean of $\delta^lz^l+\epsilon^l$" is very hard to understand.  Do you mean $\epsilon^l$ is the mean of $\delta^lz^l+\epsilon^l$?
> >
> > Finally, I'd like to note that I cannot afford to compare and check all the revisions, since they are not highlighted. But considering the unclear statements I encountered above, I am still concerning that the revised manuscript is probably not clear enough.

---

> ### Author Response · Authors · 2022-11-15
> **Reply to Reviewer Rg7n (3/4)**
>
> **Q2: It is not clear to me how the output features of the two BN layers are fused? Of if they are not fused, the model will cause the increase of computational cost in local clients and some analysis should be provided.**
>
> **A:** We apologize for this confusion. In MixIG, local feature and augmented feature are not fused, we train the network with these two features using Eq. 4 and 5, as illustrated in Fig. 2. We revised Figure 2 for clear presentation. We show the increase of computational cost of FedIG (local training with Eq. 6), especially training time cost, in Table 4. At inference time, computational cost of FedIG is same with the baseline (FedAvg) because we only use features normalized by global statistics when testing unseen clients’ data. Note that MixIG randomly interpolates instance and global statistics for feature augmentation in training. It is not used at the test time.
>
> If we use zero-shot adapter (FedIG-A), the inference cost increases because the interpolation weights should be obtained from the zero-shot adapter, as illustrated in Fig. 3. But this case also does not require two features from two BN layers. The features are extracted with estimated statistics from the zero-shot adapter by forwarding the test input only once. The zero-shot adapter estimates the interpolation weights between instance and global statistics for each BN layer as an instance-wise manner. Therefore, computational and time cost increases at every inference, as shown in Table 4.
>
> **Q3: From a unified perspective, I am wondering α in Eq.7 equivalent to μ in Eq.3, only that α are dynamically generated rather than sampling μ from a gaussian distribution. If it is this case, I would suggest to use the same symbol to ease the understanding.**
>
> **A:** Thank you for thoughtful suggestion. The forms of Eq.3 and Eq.7 are same as the reviewer mentioned, but we used different notations for interpolation parameters because their roles are totally different. We describe their roles below.
>
> For learning client-invariant representations, we use MixIG and client-agnostic learning objectives. In MixIG, we randomly interpolate instance and global statistics in Eq. 3. The interpolation weights are random values for generating diverse domains’ features. However, the interpolation weights in Eq. 7 are generated from the zero-shot adapter, and the zero-shot adapter is trained to find the proper interpolation weights at every BN layer for each input.
> In summary, $u^l$ in Eq. 3 is random interpolation values for learning client-invariant representations whereas $\alpha^l$ is the estimated values from the zero-shot adapter for normalizing the test input properly. Their roles are totally different, so we use different notation of interpolation weights for Eq. 3 and Eq. 7.

---

> ### Author Response · Authors · 2022-11-15
> **Reply to Reviewer Rg7n (2/4)**
>
> **Q1: My major concern involves with the core model design. It is not reasonable for me that the method aims to keep two groups of BN parameters at each BN layer, at the same time to let the features generated by the two parameter groups be same by Eq. 4. Would this make the BN parameters same? I am wondering how will a simplified model perform, e.g., by dropping all local BN, $L_{CE}$ in Eq. 6 and the loss in Eq. 4.**
>
> **A:** In our framework, we have two groups of BN parameters at each BN layer, i.e., local BN statistics ${\mu_k, \sigma_k}$ and global BN statistics ${\mu_G, \sigma_G}$ in Figure 2. For our client-agnostic learning, we extract two types of features (local and augmented features) and use both features to learn client-invariant representations. To get the local feature $f_{i,k}$, we forward local data to the model with local BN layers while local BN statistics are updated with batch statistics of local data (see the upper path in Fig. 2). To get the augmented feature $f_{i, \Delta}$, we forward local data to the model with randomly mixed instance and global statistics (see the lower path in Fig. 2). Here, global statistics are not updated with batch statistics of local data during local training. In summary, local BN statistics are used for learning batch statistics of local data and extracting local features and global BN statistics are used for augmenting local features. Two parameter groups have different roles in our algorithms.
>
> As the reviewer mentioned, the model is trained to build the same statistics with global statistics if we use the client-agnostic feature loss (Eq. 4) on every layer after BN layer and fix $u^l$ to 0. However, we use the client-agnostic feature loss (Eq. 4) using the last features and use randomly generated $u^l$ for feature augmentations. Also, two client-agnostic loss functions are added to the main local cross-entropy loss function for helping learn client-invariant representations while minimizing cross-entropy loss on its own dataset. If only CE loss with local and augmented features is applied, the performance is improved but its improvement is marginal as shown in Table 2 (see the performance of CACL).
>
> About a simplified model suggested by the reviewer:\
> Thanks for good suggestions, here we conducted experiments with the simplified framework. We simplified our framework by dropping the path for local features (the upper path in Fig. 2). The local models are trained with augmented features (the lower path in Fig. 2) using the cross-entropy loss function. At the test time, running mean and variance are used for normalizing the test inputs. If we use only global BN layers, we cannot update running mean and variance with training statistics because global statistics should not be changed during local training. To solve this issue, we update local BN layers with batch statistics from MixIG. Note that local BN layers only keep statistics, i.e., running mean and variance. These statistics are uploaded to the server after local training, and they are aggregated to global statistics and used at the test time. We reported the results on our simplified framework.
>
> Table 1: Performance (%) of our simplified framework
> |Method| P | A | C | S | Avg|
> |-|-|-|-|-|-|
> |FedAvg| 90.40 (0.97) | 72.52 (2.28) | 72.59 (0.37) | 73.90 (1.74) | 77.35|
> |FedBN | 92.74 (0.97) | 77.08 (0.59) | 75.08 (0.85) | 78.28 (1.14) | 80.79|
> |Our simplified framework | 92.75 (0.42) |76.83 (0.72) | 74.65 (0.01) | 80.84 (0.03) | 81.27|
> |FedIG (ours) | 92.99 (0.61) | 82.17 (1.15) | 77.71 (1.13) | 83.40 (0.19) | 84.07|
>
> Here, we do not use the zero-shot adapter in this experiment to separately analyze the impact of our model design. The performance of the simplified framework outperforms FedAvg and FedBN, but it achieves low performance compared to our proposed framework, FedIG. This result shows that our client-agnostic learning with both local and augmented features is effective on domain generalization.

---

> ### Author Response · Authors · 2022-11-15
> **Reply to Reviewer Rg7n (1/4)**
>
> Dear reviewer,
>
> Thanks for your review and thoughtful feedback. We apologize for the unclear presentation that confuses the reviewer. We first give the detailed explanation of our methods, then we clarify your concerns regarding our model design, loss function, and zero-shot adapter. It is worth noting that we revised the paper about MixIG and zero-shot adaptation for clear understanding.
>
> In this paper, we propose two strategies, one is for local training and the other one is for inference.
>
> (1) Our first approach, MixIG, is designed to learn client-invariant representations in local training without additional privacy leakage, which can be viewed as a privacy-preserving multi-source domain generalization method. The main concept of our local training is to utilize other domains’ characteristics through global statistics, i.e., aggregated statistics in the server, while keeping data private. Global statistics reflect distributions of all training clients, thus features normalized by global statistics contain relative global information. Furthermore, to generate more diverse domains’ features, we interpolate instance and global statistics with the random vector, where each element is sampled from uniform distribution in Eq. 3. Finally, we normalize local data using randomly mixed instance and global statistics to generate diverse features every iteration. With both original and augmented features as illustrated in Fig. 2, we apply our client-agnostic learning objectives, cross-entropy loss with augmented features and alignment of the original and augmented features (Eq. 4 and 5), to learn client-invariant representations.
>
> (2) Our second proposed method is zero-shot adaptation for bridging a large domain gap between training and test data directly. At the test time, we utilize instance statistics of a test input to directly bridge a domain gap as Eq. 7. Here, $\alpha^l$ is an interpolation parameter that adjusts the contribution of instance statistics of the test sample. Previous work [1, 2] uses the manually fixed value suitable to target domain, but we propose zero-shot adapter to adaptively select the proper $\alpha^l$ suitable to each test input. Zero-shot adapter is designed to interpolate the statistics based on the difference between two statistics, i.e., instance and global statistics, as illustrated in Fig. 3. The zero-shot adapter is trained through federated learning on each client and aggregated in the server. We train the zero-shot adapter with Eq. 8 in local clients, where loss function is designed to simulate the test case. After federated learning, the feature extractor, the classifier, and the zero-shot adapter are broadcasted to clients by the server. The adapter helps the feature extractor and the classifier to adapt the test input directly, which indicates zero-shot adaptation.
>
> [1] Fuming You, Jingjing Li, and Zhou Zhao. Test-time batch statistics calibration for covariate shift. arXiv preprint arXiv:2110.04065, 2021.\
> [2] Xuefeng Hu, Gokhan Uzunbas, Sirius Chen, Rui Wang, Ashish Shah, Ram Nevatia, and Ser-Nam Lim. Mixnorm: Test-time adaptation through online normalization estimation. arXiv preprint arXiv:2110.11478, 2021.

---

### Decision · Program_Chairs · 2023-01-20

**Decision:**

Reject

**Justification For Why Not Higher Score:**

According to my expertise and reviewing process, this paper should belong to a Reject.

**Justification For Why Not Lower Score:**

According to my expertise and reviewing process, this paper should belong to a Reject.

**Metareview: Summary, Strengths And Weaknesses:**

This paper studies how to learn a global model in Federated Learning (FL) from multiple distributed source domains and generalize the model to new clients in unseen domains at inference time. The authors propose client-agnostic learning with mixed instance-global statistics for local training along with zero-shot adaptation with estimated statistics for inference, which can be viewed as a privacy-preserving multi-source domain generlaization method. To obtain client-invariant representations, the authors augment local features using data distribution of other clients via aggregated BN statistics from the global model. Therefore, the paper makes a useful contribution for studying the real-world FL.

However, there are several obvious weakness: 1) The significance of novelty is limited. Particularly, as also mentioned by other reviewers, the idea of relying on consensus/confidence among two or multiple models to identify clean/noisy labels has been applied in the previous method. 2) The method is not end-to-end, and the comparison results against SOTA noisy learning method are weak. 3) It requires more computational costs for the proposed method as an ensemble of models are required. No comparison is given with methods using similar computational costs (e.g., larger models). A more careful comparison with baselines, including the computational costs, is necessary. Overall, this paper may not be ready for publication at ICLR. The next version must be a strong paper if authors can take comments into consideration.